# Influence of solution efficiency and valence of instruction on additive and subtractive solution strategies in humans, GPT-4, and GPT-4o

Lydia Uhler [1,3] ✉, Verena Jordan[1], Jürgen Buder [2], Markus Huff [1,2] & Frank Papenmeier [1]

Generative artificial intelligences, particularly Large Language Models (LLMs), increasingly influence human decision-making, making it essential to understand how cognitive biases are reproduced or amplified in these systems. Building on evidence of the human "addition bias" – a preference for additive over subtractive problem-solving strategies[1] – this research compared humans with GPT-4 (Study 1) and GPT-4o (Study 2) in spatial and linguistic tasks. Study 1 comprised four experiments (1a, 1b, 2a, 2b) with 588 human participants and 680 GPT-4 outputs; Study 2 included two experiments (3a, 3b) with 751 human participants and 1,080 GPT-4o outputs. We manipulated (a) solution efficiency and (b) instruction valence. Across both studies, a general addition bias emerged, more pronounced in the LLMs than in humans. Humans made fewer additive choices when subtraction was more efficient than addition (compared to when both were equally efficient), whereas GPT-4's output showed the opposite pattern. GPT-4o's outputs aligned with those of humans in the linguistic task but showed no efficiency effect in the spatial task. Instruction valence did not reach statistical significance for either agent in the spatial task. In the linguistic task, positive valence (compared to neutral valence) led to more additive outputs in both GPT models, but only in Study 2 for humans. These findings indicate that addition bias has been transferred to LLMs, which can replicate and, depending on context, amplify this human bias. This emphasizes the importance of further theoretical and empirical work on the cognitive and data-driven mechanisms underlying addition bias in both humans and LLMs.

*"The present letter is a very long one, simply because I had no leisure to make it shorter."*[2] (p. 417)

Research on human problem-solving increasingly examined the use of additive and subtractive solution strategies, which Adams et al. [1] systematically investigated. Problem-solving processes involve mental representations of an original state, possible transformations, and different categories of actions (e.g., addition or subtraction) that can bring about these transformations. An additive solution strategy results in the transformed state having more components than the original, whereas a subtractive strategy results in the transformed state having fewer components than the original. Adams et al. describe that people change ideas, objects, and situations with a tendency towards additive transformations, while subtractive transformations are overlooked, even if they constitute superior

solutions. This idea of *addition bias* (also referred to as *subtraction neglect*) was supported by Adams et al. in a series of studies and experiments and replicated by Fillon et al.[3] and – to a lesser extent – by Juvrud et al. [4].

Understanding the addition bias is crucial as it limits the consideration of alternative solutions and may lead to the overuse of additive strategies. The consequences are numerous, ranging from information overload[5] to policy and regulatory overload[6], the development of slow and bloated software[7], or complex computer models with increasing numbers of parameters[8].

Concurrently, Large Language Models (LLMs) such as OpenAI's GPT models attracted noticeable interest due to their performance in problem-solving tasks[9]. Nonetheless, there is a gap in the research investigating under which circumstances human problem-solving biases (e.g., *addition bias*) manifest within LLMs. We conducted two studies pursuing not only the

[1]Department of Psychology, University of Tübingen, Tübingen, Germany. [2]Leibniz-Institut für Wissensmedien, Tübingen, Germany. [3]Present address: Department of Psychology, University of Münster, Münster, Germany. ✉e-mail: lydia.uhler@uni-muenster.de

investigation of human problem-solving behavior but also the question of how biases in LLMs differ from, or align with, those observed in humans. We looked explicitly at whether GPT-4 (Study 1) and GPT-4 Omni (GPT-4o, Study 2) outputs contain more additive than subtractive transformations, in a mirror to cognitive tendencies shown by humans, or reveal patterns of behavior that are in fact unique.

## The origins and implications of addition bias

While there is good evidence for the existence of an addition bias, its mechanisms are still being investigated. On a *cognitive* level, Adams et al. [1] cite the overuse of heuristic memory search, in which cognitively accessible memory content is accessed quickly without sufficient consideration of alternatives. Various cognitive, cultural, and socio-ecological reasons may cause people to use addition more frequently, making it more accessible. Research showed that the likelihood of subtractive solution strategies increases when the possibility of subtractive transformations is explicitly cued in the instructions[1,3].

Regarding the *affective* causes of addition bias, Adams et al. [1] argue that additive changes are usually evaluated more positively. Terms such as "more" or "higher" are associated with "positive" or "better"[10]. People experience more recognition for additive transformations, whereas subtractive changes are considered less creative[11]. With the notion that subtractive change is a threat to the status quo, addition bias is in line with cognitive biases such as sunk cost fallacy[12], waste aversion[13], existence bias[14], or loss aversion[15]. Studies on the valence of instruction (positively vs. negatively connoted task goals) in relation to addition bias yielded mixed results. While Adams et al. [1] found no differences in the frequency of subtractive strategies between the task goals of improving vs. worsening the status quo, Fillon et al. [3] found that fewer subtractive solutions were generated in the improvement condition. In linguistic analyses, Winter et al. [16] showed that addition-related words are used in more positive contexts than subtraction-related words. Verbs of improvement are more often associated with addition than neutral verbs of change[16].

To better understand the conditions under which addition bias intensifies or weakens, it is necessary to examine both cognitive and affective influences. On a cognitive level, the relative efficiency of solution strategies (i.e., how many steps are required for additive vs. subtractive solutions) may affect strategic preferences: If subtraction is clearly more efficient, additive solutions may be disregarded despite default tendencies. This idea has not been systematically tested in prior work but has important implications for understanding the flexibility of human (and LLM) reasoning. On an affective level, the valence of task instructions may influence agents to display additive strategies, as goals with positive connotations (e.g., "optimize" or "enhance") may implicitly cue the desirability of adding rather than removing elements. Prior findings have been mixed[1,3], and the contribution of affective framing to problem-solving behavior remains insufficiently understood.

Since the initial investigation of addition bias, studies showed that its effects are anchored in various domains that extend far beyond the empirical evidence and the illustrative real-world examples presented by Adams et al. [1]. The general disregard for subtractive solutions and the value of such manifest in a healthcare system that focuses exclusively on prescribing and developing more drugs (vs. on contemplating what could be removed or what behaviors could be stopped to make diseases disappear)[17], in the erroneous perception of recycling (vs. reducing waste generation) as the most sustainable waste management measure[18] or in the significantly higher likelihood of longer novels (vs. shorter novels on shortlists) to win literary awards[19]. In view of such far-reaching effects, the question arises whether generative artificial intelligence (AI) makes more rational decisions and thus generates superior solutions to human problems.

## Large language models

With the introduction of *Chat Generative Pre-Trained Transformer* (ChatGPT) in 2022, the use of and the debate about generative AI entered the public sphere. Applications range from software development and

testing to writing poems, essays, business letters, or contracts[20]. The general discussion focuses on LMMs that use deep learning for *Natural Language Processing* (NLP) and are trained to predict the next token, i.e., individual units of text. OpenAI's GPT models (e.g. GPT-4 or GPT-4o) are among the most powerful LLMs of the current generations[21], although numerous other models are now available from various research organizations and companies, e.g. PaLM by Google AI, LLaMA by Meta AI or Luminous by Aleph Alpha. In contrast to its predecessors (e.g., GPT-3.5), GPT-4 and GPT-4o are large-scale, *multimodal* models that can process text as well as images.

## Performance and biases of LLMs: previous psychological research

As AI systems become increasingly powerful, a lack of transparency persists regarding the internal training processes and inference mechanisms of these agents[9,22]. This opacity has increased concern about how these systems might mirror human cognitive and social patterns. Osborne et al. [23] identify several "entry points" through which human regularities and biases can influence AI models, including (1) biased or unrepresentative training data and human annotations embedding social and linguistic regularities, (2) design and parameter choices shaped by model engineers' implicit biases, and (3) insufficient awareness or negligence regarding fairness during model development. Beyond these bias transmission pathways, corpus-linguistic evidence indicates that written language itself disproportionately represents additive transformations. Winter et al. [16] showed that GPT-3's cloze probabilities are systematically biased toward additive continuations and that verbs of change and improvement exhibit stronger semantic associations with addition than with subtraction in embedding space. Frequency analyses of large reference corpora further demonstrate that words denoting increases (e.g., "add", "more") occur more often than those denoting decreases (e.g., "subtract", "less") in English, with comparable asymmetries reported for German[24].

Researchers proposed using psychological methods to better understand the processes of the "black box" (see ref. 25,26). Cognitive psychology, in particular, offers valuable tools for evaluating LLMs' performance and identifying biases similar to those observed in humans (see refs. 9,27,28).

Several studies were conducted to systematically analyze the capabilities of LLMs using cognitive psychological methods. Early research concluded that GPT-3's performance fell behind human abilities, for example, in inductive reasoning[29] and causal hypothesis formation[30]. By contrast, Webb et al. [31] found that GPT-3 solved analogy tasks with performance levels matching or exceeding those of human samples. Binz and Schulz[9] investigated GPT-3' performance in decision making, information search, deliberation, and causal reasoning. While no human abilities were recognizable in areas such as causal reasoning, GPT-3 achieved human-like or even better performance in other experiments. The authors point out that the well-known tasks were likely part of the training data for GPT-3. The answers could have been based on explicit knowledge of these experiments instead of actual abilities. Besides investigating performance, Binz and Schulz explored six cognitive biases presented by Kahneman and Tversky[32] and found that GPT-3 mirrored three of them (framing effect, certainty effect, and overweighting bias).

With the increasing size of subsequent models, studies found improvements in performance. The analogical reasoning abilities exhibited by GPT-3 increased in GPT-4[31]. Dhingra et al. [33] evaluated GPT-4's capabilities on a set of cognitive psychology datasets and found high levels of accuracy in different tasks. In a battery of semantic illusions and cognitive reflection tests designed to evoke biased (and incorrect) responses, both GPT-3.5 and GPT-4 outperformed humans by generating hyperrational responses, whereas earlier LLMs were unable to match human performance[34]. This steep shift in rationality does not hold true for other biases. Suri et al. [35] demonstrated the prevalence of anchoring heuristics, representativeness heuristics, availability heuristics, framing effects, and endowment effects in GPT 3.5. Similarly, Yax et al. [28] found that reasoning errors often attributed to heuristic-based human thinking also appeared in LLMs.

Shahriar et al. [36] evaluated GPT-4o across language, vision, speech, and multimodal domains, comparing its performance with that of its predecessors GPT-3.5 and GPT-4. While GPT-4o processed multimodal inputs faster and more efficiently than GPT-4, this was partially achieved at the expense of accuracy in complex tasks (e.g., USMLE Step 1: 83.1% accuracy for GPT-4o vs. 90.0% for GPT-4). The model achieved high scores in language and reasoning tasks but showed greater variability in vision and audio evaluations. Findings suggest that the outputs depend strongly on question wording, the order in which information is presented, and the presence of unclear or contradictory information.

## The present studies

The present research intended to further explore problem-solving processes, specifically the use of additive and subtractive solution strategies by humans and GPT-4 (Study 1) and GPT-4o (Study 2), which led to three objectives:

(1) to investigate human addition bias and to replicate previous research yielding inconsistent results,
(2) to extend the research of addition bias to current GPT models, and
(3) to explore differences between humans and current GPT models in choosing additive or subtractive strategies.

Thereby, this investigation enabled a deeper understanding of the cognitive basis of human problem-solving. If GPT-4 or GPT-4o and humans display similar results, it may suggest that the technological foundations of LLMs closely mimic human cognition. However, if distinct differences emerge, it may highlight that human cognition involves more than just processing words, as LLMs do, suggesting a more complex, nuanced approach to problem-solving in humans.

While spatial capabilities of GPT-4 improved compared to earlier models, its performance on such tasks is still highly variable[37]. Xu et al. [38] introduced a multi-task spatial benchmark to evaluate LLMs across a variety of spatial reasoning tasks, like simple route planning in grid-based simulated environments. Findings show that GPT-4o achieved the highest overall accuracy (71.3% in the first phase). We therefore decided to use two tasks – one spatial (*symmetry task*) and one linguistic (*summary task*) – to be able to draw conclusions based on the problem type.

The presence and magnitude of addition bias may be influenced by both cognitive and affective factors. To shed further light on these processes, we included two task variations. On the cognitive level, we varied *solution efficiency* in two conditions: one in which the use of additive and subtractive solution strategies was equally efficient (*addition and subtraction equally efficient* condition) and one in which the use of a subtractive strategy was more efficient than the use of an additive strategy (*subtraction more efficient* condition). On the affective level, we varied the *valence of instruction* by including one condition in which the valence of the instructive verb of the task was neutral (*valence neutral* condition) and one in which it was positive (*valence positive* condition). Drawing on the results of previous research, we proposed three hypotheses for both humans and GPT-4 (Study 1) or GPT-4o (Study 2):

> **H1 (Addition Bias):** *Additive solution strategies are chosen significantly more often than subtractive solution strategies across all conditions.*

> **H2 (Solution Efficiency):** *For both spatial and linguistic problems, the likelihood of choosing additive solution strategies is larger when the solution efficiency for addition and subtraction is equal than when the solution efficiency for subtraction is higher.*

> **H3 (Valence of Instruction):** *For both spatial and linguistic problems, the likelihood of choosing additive solution strategies is larger when the valence of the instruction is positive than when the valence of the instruction is neutral.*

Regarding the third research objective, which investigates potential differences in the problem-solving behavior of humans and GPT models,

previous research did not provide a basis for hypotheses. Studies confirmed highly accurate and rational performances of GPT-4 (see refs. 33,34), which could lead to the assumption that addition bias is less prevalent in the LLM than in humans. Winter et al. [16], on the other hand, showed that addition bias was not only manifested in behavioral choices but also in language itself. The analyses were extended to GPT-3 revealing that the LLM adopted more positive connotations with addition-related as compared to subtraction-related words. GPT-3 predicted the word "adding" to follow verbs of change (and even more so verbs of improvement) significantly more often than the word "removing". Thus, these results suggest that a human-like addition bias might also be present in subsequent GPT models. Hence, for the present study, the exploration of differences between humans and GPT models in choosing additive or subtractive strategies was included as an additional, open research question:

> **Research Question (RQ):** *Do GPT-4 / GPT-4o and humans use additive and subtractive solution strategies in the same way?*

In Study 1, we conducted a series of four experiments, combining both problem types and both variations. In Experiments 1a and 1b, we varied solution efficiency: in the symmetry task for Experiment 1a and in the summary task for Experiment 1b. The valence of instruction was kept constant (neutral). In Experiments 2a and 2b, we varied the valence of instruction while solution efficiency was kept constant (addition and subtraction equally efficient). Experiment 2a focused on the symmetry task, Experiment 2b on the summary task. Study 2 was designed to extend and refine Study 1 by replacing GPT-4 with GPT-4o and by addressing methodological constraints identified in Study 1. Specifically, Study 2 employed a fully crossed design that manipulated solution efficiency and instruction valence simultaneously across both tasks, enabling direct tests of their interaction.

## Methodology
### Openness and transparency

All hypotheses, research questions, methods, analyses, and exclusion criteria for Studies 1 and 2 were preregistered. Any deviations from the preregistrations are explicitly marked in the respective sections. Preregistrations are available on the Open Science Framework (OSF): https://osf.io/6pkwb (Study 1, submitted: August 2, 2023, updated: August 9, 2023) and https://osf.io/7kh2b (Study 2, submitted: March 21, 2025). Data and analysis scripts are provided: https://osf.io/c78rm/[39].

### Participants and data generation

In both studies, human participants were recruited via Prolific, restricted to individuals based in the United States who are aged 18 or older and have high English proficiency, using personal computers.

To attain 80% power in Study 1, we aimed for 170 human participants and an equivalent number of GPT-4 iterations for each of the four experiments. Detailed information on the power calculations is stated in the preregistration of Study 1. GPT-4 iterations were generated via the OpenAI API[40] with a temperature setting of 0.7 (default). Both agents completed identical tasks under controlled conditions for direct comparison of human and LLM outputs. To attain 80% power in Study 2, we aimed for 540 human participants per experiment (135 per condition across four conditions) and an equivalent number of GPT-4o iterations. This target was set separately for each of the two experiments (Experiments 3a and 3b). Detailed information on the power calculations can be found in the preregistration of Study 2. GPT-4o iterations were generated using the OpenAI API[40] with the temperature parameter set to 1.0 (default; in contrast to 0.7 used in Study 1).

To ensure the comprehensibility of the linguistic symmetry task (for more details, see section Materials), a test run was carried out prior to conducting Study 1. To address issues with externally generated responses (e.g., via ChatGPT, see ref. 41), the preregistration of Study 1 was updated to exclude participants whose web page lost focus during the task (via the *window.blur event* method). In Study 2, two additional safeguards were

introduced to further ensure data quality and prevent externally assisted or automated responses: a hidden attention-check item in white font and Google reCAPTCHA v2. These measures were added in response to growing concerns about automated submissions[42,43]. It should be noted that they may have inadvertently excluded some human participants, e.g., those using screen readers, dark mode, or those unable to solve the Captcha.

In Study 1, data collection proceeded in two steps. As the first step, data was collected until 170 participants (85 participants per condition) finished each experiment. Preliminary data filtering was carried out so that duplicates and externally assisted responses were eliminated as preregistered. Participants who did not provide informed consent or did not agree to the usage of their data were also excluded. In a second step, the retained data was analyzed for usability. Answers were excluded in case the solution entailed a final filled field count (symmetry task) or final word count (summary task) that equaled the original count of the respective problem (for details, see section Procedure and Measures). After data exclusion, a second round of data collection was conducted to replace excluded participants and achieve the targeted sample sizes. The same data exclusion criteria were applied. Due to unforeseeably high exclusion rates in the symmetry task (for details, see section Limitations), targeted sample sizes were not reached after the second round of data collection. For economic reasons, no further data was collected. Participants received £0.50 for participation in the symmetry task and £1.00 in the summary task. The Ethics Committee of the Leibniz-Institut für Wissensmedien approved the research protocol covering both Study 1 and Study 2 (LEK 2023/043).

Data from 170 GPT-4 iterations per experiment (85 iterations per condition) were collected and subsequently filtered to exclude incomplete answers. Incompletion was attributed to having reached the maximum number of tokens specified. Two cases were excluded for this reason. Analogous to the human data, the remaining answers were analyzed and excluded if they could not be evaluated (i.e., unchanged filled field or word counts after transformation). In a second round of data collection, another 170 iterations per experiment were conducted. The additional answers were successively analyzed and added to the final samples until the targeted sample sizes were reached. The remaining iterations were not considered further for the study.

Sample sizes of the human and GPT-4 samples before and after data exclusion, as well as the demographic characteristics of the final human samples, are shown in Table 1. Data on race or ethnicity were not collected in this study. A detailed breakdown of data exclusions is provided in Tables A1 and A2 in Supplementary Information (SI 1).

In contrast to Study 1, data collection of human participants in Study 2 did not follow a two-step procedure. Instead, the initial sample size was increased in advance based on the exclusion rates observed in Study 1, aiming to reach the targeted number of valid cases within a single round of data collection. After data collection, the preregistered exclusion criteria were applied to filter out invalid responses, using the same criteria as in Study 1. Participants received £0.50 for completing the symmetry task and £1.00 for the summary task. Due to higher-than-anticipated exclusion rates, particularly in the spatial task, the final sample size deviated from the targeted sample size. No further data collection was conducted.

For GPT-4o, 540 iterations per experiment (135 per condition) were generated in a first round and filtered for completeness. Responses that were incomplete due to token limits were excluded. This did not occur in Study 2; therefore, no exclusions were made on this basis. For the symmetry task, each human answer and each GPT-4o trial was deemed valid only if (a) the final colored-box count differed from the original pattern, as in Study 1, and additionally for GPT-4o trials if (b) both output formats (text-based and grid-based) were presented within a single iteration (as was the case for all but one iteration). All retained outputs were independently coded by two raters, with discrepancies resolved by discussion (Cohen's κ reported in Procedure and measures). A second batch of GPT-4o iterations was then collected; valid cases were coded and added sequentially until the target of 135 valid trials per condition was reached.

Sample sizes of the human and GPT-4o samples before and after exclusion, along with demographic characteristics of the final human

**Table 1 | Summary of sample sizes and demographic characteristics (Experiments 1a, 1b, 2a, 2b)**

| Experiment / Condition | Human samples | | | | GPT-4 Samples | |
|---|---|---|---|---|---|---|
| | Sample sizes (n) | | Demographics[a] | | Sample sizes (n) | |
| | Before exclusion | Final sample | Gender (n) (male / female / non-binary)[b] | Age (years) (M / SD) | Before exclusion | Final sample |
| Experiment 1a | | | | | | |
| add and sub equally eff | 191 | 67 | 33 / 32 / 2 | 33.40 / 10.89 | 107 | 85 |
| sub more eff | 124 | 65 | 30 / 31 / 4 | 34.54 / 10.97 | 94 | 85 |
| Total | 315 | 132 | 63 / 63 / 6 | 33.96 / 10.90 | 201 | 170 |
| Experiment 1b | | | | | | |
| add and sub equally eff | 107 | 87 | 39 / 45 / 2 | 36.90 / 12.30 | 91 | 85 |
| sub more eff | 99 | 84 | 40 / 43 / 1 | 37.58 / 12.41 | 86 | 85 |
| Total | 206 | 171 | 79 / 88 / 3 | 37.23 / 12.32 | 177 | 170 |
| Experiment 2a | | | | | | |
| val neutral | 178 | 60 | 31 / 27 / 2 | 35.48 / 10.67 | 111 | 85 |
| val positive | 174 | 59 | 34 / 23 / 2 | 34.69 / 10.96 | 111 | 85 |
| Total | 352 | 119 | 65 / 50 / 4 | 35.09 / 10.77 | 222 | 170 |
| Experiment 2b | | | | | | |
| val neutral | 107 | 83 | 43 / 37 / 2 | 37.59 / 11.60 | 89 | 85 |
| val positive | 121 | 83 | 36 / 46 / 1 | 36.43 / 12.39 | 86 | 85 |
| Total | 228 | 166 | 79 / 83 / 3 | 37.01 / 11.98 | 175 | 170 |

*Note.* add and sub equally eff = addition and subtraction equally efficient; sub more eff = subtraction more efficient (than addition); val neutral = valence of instruction neutral; val positive = valence of instruction positive.
[a]Final samples only.
[b]Deviations from final sample sizes are due to some participants' nondisclosure of gender (in Experiment 1b, one participant did not disclose their gender in the "add and sub equally eff" condition; in Experiment 2b, one participant did not disclose their gender in the "val neutral" condition).

samples, are presented in Table 2. Data on race or ethnicity were not collected in this study. A detailed breakdown of data exclusions is provided in Tables A3-A6 in Supplementary Information (SI 2).

## Materials

Both tasks in Studies 1 and 2 are adapted versions of those in Adams et al. [1]. The *symmetry task* is a linguistic version of their grid task, which represents the *spatial* problem in Studies 1 and 2. Participants were presented with a grid with a non-symmetrical field pattern. A field could either be filled ("[X]") or empty ("[]"). They were then asked to describe how to change the pattern to make it "perfectly symmetrical from left to right and from bottom to top" by switching fields from "[X]" to "[ ]" or from "[ ]" to "[X]" with as little switching as possible. A small grid (4-by-4) was used to account for the increased complexity of solving this task by articulating the proposed solution instead of interactively clicking on the grid. Note that prior to conducting Study 1, the test run of the task revealed that stating the task as "toggle the color of any box" was not comprehensible for most participants, which is why we replaced it with the above-described "switch" instructions for Study 1.

In Experiment 1a (Study 1), the *solution efficiency* variation was implemented by either presenting an initial field pattern for which the most efficient additive and subtractive solutions required the same amount of switching (addition and subtraction equally efficient condition) or an initial field pattern for which the most efficient solution was subtraction (subtraction more efficient condition). In Experiment 2a (Study 1), the *valence of instruction* was varied by exchanging the instructive verb of the task from "change" (valence neutral condition) to "improve" (valence positive condition). Figure 1 illustrates the different variations used in the symmetry task.

The *summary task* constitutes the *linguistic* problem of Studies 1 and 2. Participants were provided with a newspaper article and a corresponding summary and asked to edit the summary. While the task itself was very similar to the one presented by Adams et al. [1], a different, publicly available text (see ref. 44) and summary (see ref. 45) were used. In Experiment 1b (Study 1), *solution efficiency* was varied by providing

the word count of the initial summary and by specifying word count ranges that were prohibited for the edited summaries. While the prohibited range in the addition and subtraction equally efficient condition was ±12 words around the original word count, the subtraction more efficient condition was operationalized by shifting the range so that getting outside the range required subtracting at least six words or adding at least 18 words. In Experiment 2b (Study 1), the variation of valence of instruction was implemented by exchanging the instructive verb "edit" used in the neutral condition with the verb "improve" in the positive condition. An overview of the variations used in the summary task is provided in Fig. 2. The materials of Experiments 1a, 1b, 2a, 2b (all Study 1) can be found in the Supplementary Information (SI 3).

As noted, Study 2 used the same two tasks as Study 1 (symmetry, summary). Stimuli and evaluation criteria were identical unless otherwise specified. The four experimental cells, i.e., solution efficiency (addition and subtraction equally efficient vs. subtraction more efficient) x valence of instruction (neutral vs. positive), were realized for both tasks; see Fig. 3 (symmetry) and Fig. 4 (summary).

In Experiment 3a (Study 2), participants and GPT-4o again received a $4 \times 4$ grid and were asked to describe how to obtain perfect symmetry. To minimize unintended interpretations and raise accuracy, the instructions explicitly stated not to move the "X", and the action verb was updated from "switch" to the more precise "toggle". As in Study 1, efficiency conditions were implemented via initial field patterns and valence of instruction was manipulated via the goal verbs ("change" vs. "improve"). To strengthen the manipulation, the goal verb was included four times in the instructions, compared to once in the symmetry task and twice in the summary task in Study 1. Once again, there were no restrictions on how participants or the model presented their final solution in the open text field. In the symmetry task, responses could be given in text-based or grid-based formats, or both, with all formats being retained for subsequent coding. In practice, human participants usually produced only one response format, whereas GPT-4o almost always generated both formats within the same output (see Section 5.4 Procedures and measures and Table A5 in the Supplementary Information SI 2). This approach ensured that both response formats were

## Table 2 | Summary of sample sizes and demographic characteristics (Experiments 3a and 3b)

| Experiment / Condition | Human samples | | | | GPT-4o Samples | |
|---|---|---|---|---|---|---|
| | Sample sizes (*n*) | | Demographics[a] | | Sample sizes (*n*) | |
| | Before exclusion | Final sample | Gender (*n*) (male / female / non-binary)[b] | Age (years) (*M / SD*) | Before exclusion | Final sample |
| Experiment 3a | | | | | | |
| add and sub equally eff + val neutral | 193 | 66 | 30 / 35 / 1 | 37.91 / 11.00 | 160 | 135 |
| sub more eff + val neutral | 188 | 95 | 54 / 40 / 1 | 39.81 / 12.09 | 155 | 135 |
| add and sub equally eff + val positive | 194 | 57 | 32 / 23 / 2 | 39.16 / 10.91 | 165 | 135 |
| sub more eff + val positive | 197 | 91 | 43 / 45 / 2 | 40.08 / 12.12 | 145 | 135 |
| Total | 772 | 309 | 159 / 143 / 6 | 39.36 / 11.63 | 625 | 540 |
| Experiment 3b | | | | | | |
| add and sub equally eff + val neutral | 194 | 102 | 49 / 53 / 0 | 41.88 / 13.03 | 137 | 135 |
| sub more eff + val neutral | 198 | 107 | 53 / 53 / 1 | 43.76 / 13.14 | 137 | 135 |
| add and sub equally eff + val positive | 189 | 114 | 41 / 71 / 2 | 40.52 / 13.60 | 136 | 135 |
| sub more eff + val positive | 195 | 119 | 60 / 59 / 0 | 42.45 / 12.43 | 135 | 135 |
| Total | 776 | 442 | 203 / 236 / 3 | 42.14 / 13.06 | 545 | 540 |

*Note*. add and sub equally eff = addition and subtraction equally efficient; sub more eff = subtraction more efficient (than addition); val neutral = valence of instruction neutral; val positive = valence of instruction positive.
[a]Final samples only.
[b]Deviations from final sample sizes are due to some participants' nondisclosure of gender (in Experiment 3a, one participant did not disclose their gender in the "sub more eff + val positive" condition).

**Fig. 1 | Overview of variations used in the symmetry task (Experiments 1a and 2a).** *Note.* The red circle and the red font serve illustrative purposes only and were not present in the actual experiments. Complete materials for Study 1 are available in Supplementary Information SI 3.

**Fig. 2 | Overview of variations used in the summary task (Experiments 1b and 2b).** *Note.* The red font serves illustrative purposes only and was not present in the actual experiments. Complete materials for Study 1 are available in Supplementary Information SI 3.

available for analysis, while keeping the same task prompt structure as in Study 1.

In Experiment 3b (Study 2), agents were provided with the same base article and summary as in Study 1. The neutral instruction verb was modified from "edit" to "change" to promote consistency between the two tasks; the positive condition maintained the use of "improve". As in the symmetry task, the goal verbs occurred four times per item. The manipulation of solution efficiency via the word-count windows matched Study 1. The materials for Experiments 3a and 3b are available in Supplementary Information (SI 4).

**Procedure and measures**

In Study 1, for the human sample, a survey was constructed containing questions regarding the participants' gender and age as well as the experiment-specific task in one of the two conditions (between-subject design). After providing informed consent, participants were randomly assigned to one condition. Each participant was allowed to participate in only one of the four experiments. Each human participant and each GPT-4 iteration contributed one response to an experiment. Thus, the dataset contains one observation per case. Experiments were conducted sequentially, allowing exclusion of previous participants for subsequent

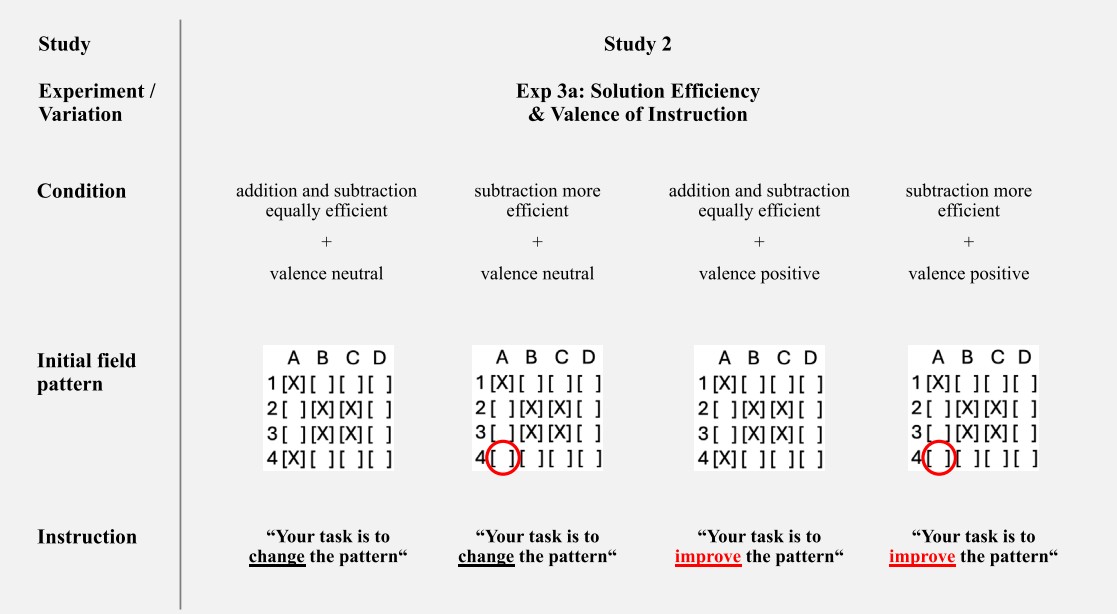

**Fig. 3 | Overview of variations used in the symmetry task (Experiment 3a).** *Note.* The red circles and the red font serve illustrative purposes only and were not present in the actual experiments. For each condition, the goal verbs ("change" or "improve") appeared four times in the instructions. Only one instance is shown here for illustration purposes. Complete materials for Study 2 are available in Supplementary Information SI 4.

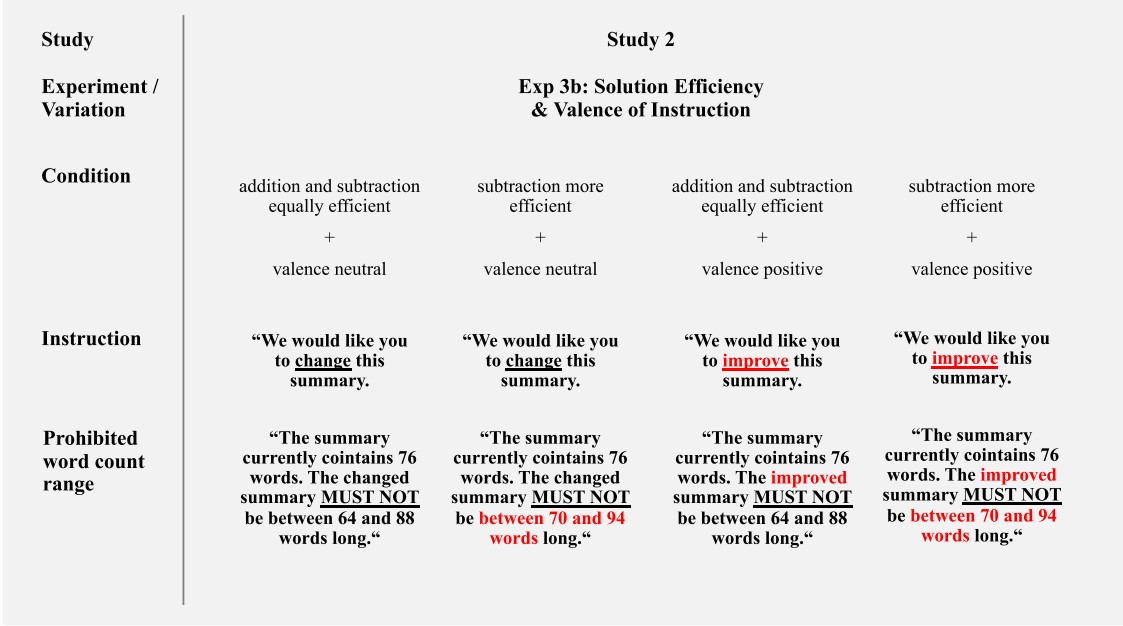

**Fig. 4 | Overview of variations used in the summary task (Experiment 3b).** *Note.* The red font serves illustrative purposes only and were not present in the actual experiments. For each condition, the goal verbs ("change" or "improve") appeared four times in the instructions. Only one instance is shown here for illustration purposes. Complete materials for Study 2 are available in Supplementary Information SI 4.

experiments. Before completing the assigned task, participants were instructed neither to leave the website nor to use external help to solve the problem. After completing the survey, participants received a debriefing on the study's purposes and were then asked to give their final consent to data usage. GPT-4 iterations were generated using the OpenAI API[40], the temperature parameter at 0.7, and the maximum number of tokens set to 500. GPT-4 was instructed to be "a helpful assistant" (default).

After data collection for Study 1, the data of Experiments 1b and 2b (summary task) were analyzed and coded via R. For Experiments 1a and 2a (symmetry task), an automated evaluation of the answers was not possible. Analysis and coding were performed manually by one of the study authors. GPT-4 displayed most answers in both a text-based and a grid-based response format. As it is an LLM trained on text, we decided to only code for text-based responses. The retained solutions in the final samples were categorized as either additive or subtractive solution strategies, regardless of the accuracy of the answer given regarding the respective task. For the symmetry task, a solution strategy was considered additive (vs. subtractive) if the final filled field count after the proposed transformation was higher (vs.

lower) than the original filled field count. Correspondingly, in the summary task, a higher (vs. lower) final word count than the original was regarded as additive (vs. subtractive). As preregistered, the accuracy of the solutions was evaluated separately for additional analyses on correct solutions only. Answers were considered as accurate if the proposed solution resulted in a grid pattern that was symmetrical from left to right and from top to bottom (symmetry task) or if the word count of the edited text was outside the specified prohibited ranges (summary task).

In Study 2, for the human sample, a survey was constructed including questions on participants' gender and age, followed by the experiment-specific task in one of the four experimental conditions (between-subjects design). After providing informed consent, participants were randomly assigned to one condition. Each participant was allowed to take part in only one of the two experiments in Study 2. The experiments were conducted sequentially, enabling the exclusion of participants from previous experiments in the recruitment for subsequent experiments. Prior to task completion, participants were instructed not to leave the study website and to refrain from using any external assistance. Upon completion, they received a debriefing about the study's purpose and were asked for their final consent to data usage. GPT-4o iterations were generated via the OpenAI API[40] using a temperature setting of 1.0 and a maximum token limit set to 4096 tokens. The system message followed the default instruction to behave as a "helpful assistant".

In Experiment 3b (Study 2, summary task), responses were processed in R. As in Study 1, solutions were classified as additive when the final word count exceeded the original and subtractive when it was lower; accuracy was defined as a final word count outside the prohibited range as specified by the solution efficiency condition.

In Experiment 3a (Study 2, symmetry task), two independent raters coded each response for (a) the final number of filled fields and (b) solution accuracy, each recorded separately for the two response formats (text-based and grid-based). They also coded (c) alignment across response formats, i.e. whether the text response, when translated into a grid, matched the provided grid, and (d) special cases (multiple solutions; attempts to "move" an "X" rather than toggle; claims that the task was unsolvable; unclear or nonsensical answers). Inter-rater reliability was computed overall, pooled across both agents (humans, GPT-4o) and all four conditions, yielding Cohen's $\kappa = .911$, with 92.75% agreement ($N = 4.428$). Disagreements were resolved through discussion, with a third rater overseeing cases that could not be determined. For human responses, the vast majority (543/545; 99.63%) exhibited a single response format, either text-based (527/543) or grid-based (16/543), while a minority (2/545; 0.37%) displayed both. Cases with inconsistent outcomes between the text and grid were excluded; in this instance, both cases were removed from the analyses (see Table A3 Supplementary Information SI 2). For GPT-4o, most trials contained both output formats within a single trial (624/625; 99.84%). If there was only one format within a single trial, the respective trial was excluded. This was the case for one single trial (1/625; 0.16%; see Table A5 Supplementary Information SI 2). Within the final sample ($4 \times 135 = 540$ trials), the final box counts across formats only matched in 31.30% of trials (169/540), while the strategy label (additive vs. subtractive), representing the study's primary outcome, coincided in 97.96% (529/540). For Experiment 3a (Study 2, symmetry task), all inferential analyses for humans were based on the prioritized text-based response format or the grid-based response format when no text response was provided. Inferential analyses for GPT-4o were conducted separately for both output formats. Given the high agreement in strategy labels, the results reported in the main text refer to the text-based outputs, as results based on grid-based solutions showed the same pattern.

### Analytic strategy

All data analyses were performed using R. All tests were two-tailed with an alpha level of .05. Testing the addition bias across all conditions (H1), a chi-square test was performed to determine whether the total frequency of additive solution strategies significantly differed from the total frequency of subtractive strategies for all solutions regardless of their accuracy. A chi-

square test of homogeneity for the 2 (agent: human vs. GPT-4 [Study 1]; agent: human vs. GPT-4o [Study 2]) x 2 (frequency of solution strategy: additive vs. subtractive) contingency table was calculated to investigate whether humans and GPT-4 or GPT-4o use additive and subtractive strategies in the same manner (RQ). The corresponding Phi coefficients were calculated as a measure of effect size.

Logistic regression analyses were conducted separately for each experiment to explain the choice of solution strategies (additive vs. subtractive). Type-II ANOVAs were calculated using the *Anova()* function in the *car* package[46]. Coefficients for the logistic regression analyses are provided in the Supplementary Information (SI 5 for Study 1, SI 6 for Study 2). Analyses were performed on all retained answers of the final samples, including those that presented incorrect solutions to the respective problem. Chi-square tests and logistic regression analyses were chosen due to categorical dependent variables (frequency of solution strategy: additive vs. subtractive).

In Study 1, to test whether the likelihood of choosing additive solution strategies was larger when the solution efficiency for addition and subtraction was equal, as compared to the solution efficiency being higher for subtraction (H2), binomial generalized linear models with logit link function were calculated for Experiments 1a and 1b. These models included the main effects of both agent (human vs. GPT-4) and solution efficiency (addition and subtraction equally efficient vs. subtraction more efficient) as well as their interaction as predictors. In the same manner, binomial generalized linear models including the main effects of agent (human vs. GPT-4) and valence of instruction (neutral vs. positive) as well as their interaction were computed for Experiments 2a and 2b to analyze whether the likelihood of choosing additive solution strategies was larger when the valence of the instruction was positive as compared to neutral (H3). In case of significant interaction effects, post hoc simple main effects were calculated for humans and GPT-4 separately to explore the agents' respective patterns of solution strategies chosen in the different conditions (RQ).

In Study 2, to test hypotheses H2 and H3, logistic regression analyses were conducted separately for each experiment. Binomial generalized linear models with a logit link function were calculated to predict the binary outcome variable (solution strategy: additive vs. subtractive). Each model included the main effects of agent (human vs. GPT-4o), solution efficiency (addition and subtraction equally efficient vs. subtraction more efficient), and valence of instruction (neutral vs. positive), as well as all two-way interactions and the three-way interaction between these predictors. Like in Study 1, Type-II ANOVAs were computed using the *Anova()* function from the *car* package[46]. In cases of significant interaction effects, planned post hoc analyses were conducted, including follow-up two-way interactions (if a three-way interaction was significant) and simple main effects to explore agent-specific patterns of strategy selection (RQ).

All models, comparisons, and post hoc analyses were pre-registered, following the experimental design and specific directional hypotheses. Therefore, no correction for multiple testing was used. To evaluate whether solution accuracy affects the types of solution strategies chosen (i.e., well-thought-out solutions being less prone to biases, see ref. 47), all analyses were planned to be conducted on subsamples consisting solely of accurate solutions as pre-registered.

## Results

The following results are reported separately for Study 1 and Study 2 to document baseline effects and their subsequent replication and extension under the refined experimental design.

### Study 1

**Hypothesis 1: cross-sample presence of addition bias.** The chi-square test revealed significant differences in the total frequencies of chosen solution strategies, with additive strategies being generated more frequently than subtractive ones across all conditions (see Table 3; $\Phi = 0.29$, 95% CI [0.24, 0.34]). This finding indicates a robust addition bias. A subsequent exploratory chi-square test of homogeneity demonstrated

that the distribution of chosen solution strategies differed significantly between agents (see Table 4; Φ = 0.14, 95% CI [0.09, 0.20]). Both human participants and GPT-4 produced more additive than subtractive strategies. The share of additive strategies was higher for GPT-4 than for human participants. In both tests, the expected frequencies were found to be ≥ 5, thereby satisfying the underlying assumptions of the tests.

**Hypothesis 2: impact of solution efficiency on choice of additive solution strategy.** The results of the logistic regression analysis for Experiment 1a (symmetry task) and Experiment 1b (summary task) are outlined in Table 5. In both experiments, the main effect of agent and the interaction between solution efficiency and agent were found to be significant. In contrast, the main effect of solution efficiency alone was not.

The results of the post hoc analysis of the simple main effects are presented in Table 6. In Experiment 1a, post hoc analyses indicated that the relationship between solution efficiency and the choice of additive strategies was significant but reversed for humans and GPT-4. In accordance with the hypothesis, human participants chose additive strategies significantly more often in scenarios where the efficiency of addition and subtraction were equal, as opposed to scenarios where subtraction exhibited higher efficiency in comparison to addition. GPT-4 exhibited the opposite outcome pattern, generating additive strategies with greater frequency in the "subtraction more efficient" condition in comparison to the "addition and subtraction equally efficient" condition. In Experiment 1b, humans and GPT-4 again exhibited significant but opposing patterns, analogous to those observed in Experiment 1a.

**Hypothesis 3: impact of valence of instruction on choice of additive solution strategy.** The results of the logistic regression analysis for Experiments 2a (symmetry task) and 2b (summary task) are presented in Table 7. In Experiment 2a, neither the main effect of valence of instruction nor the main effect of agent, nor their interaction, reached significance. In Experiment 2b, significant main effects of agent and valence of instruction were observed, as well as a significant interaction between valence of instruction and agent.

The results of the post hoc analyses of main effects are summarized in Table 8. For Experiment 2a, post hoc analyses are reported solely for completeness. As neither the main effects nor the interaction reached significance, these results are not meaningful for the purpose of inference. No statistically reliable differences in the choice of additive solution strategies were observed between the neutral and positive valence conditions for either humans or GPT-4. In Experiment 2b, the association between the valence of instruction and the likelihood of choosing additive strategies was significant for GPT-4 but not for human participants. GPT-4 exhibited a significantly higher propensity to employ additive strategies in response to positive instructions compared to neutral instructions, whereas no statistically reliable effect of valence on human strategy choice was observed.

**Additional analyses: accuracy of solutions.** As preregistered, all analyses were additionally planned to be conducted on the subsamples of solely accurate solutions. However, the evaluation of solution accuracy revealed that the distribution of accurate and inaccurate solutions differed tremendously between humans and GPT-4 (see Table 9). While the share of accurate solutions proposed by humans was at least 80% in each experiment, GPT-4 produced hardly any accurate solutions in the symmetry task (Experiments 1a and 1b) and mainly shares below 20% in the summary task (Experiments 1b and 2b). Consequently, separate analyses solely for accurate solutions were not feasible.

## Study 2
**Hypothesis 1: cross-sample presence of addition bias.** The chi-square test revealed a significant difference in the frequencies of chosen solution strategies, with additive strategies being selected more frequently than subtractive ones across all conditions (see Table 10; Φ = 0.58, 95% CI [0.53, 0.63]). This finding suggests the presence of a robust addition bias. The subsequent chi-square test of homogeneity showed that solution strategies differed significantly between agents (see Table 11; Φ = 0.51, 95% CI [0.46, 0.55]). Both samples showed a preference for additive solution strategies. The proportion of additive strategies was higher for GPT-4o than for humans. In both tests, the expected frequencies were ≥ 5, thereby satisfying the underlying assumptions.

### Table 3 | Frequencies and chi-square results for chosen solution strategy

| Solution Strategy Chosen | | N | $\chi^2(1)$ | p |
|---|---|---|---|---|
| **Additive** | **Subtractive** | | | |
| 816 (64.35%) | 452 (35.65%) | 1268 | 104.49 | <0.001 |

### Table 4 | Frequencies and chi-square results for chosen solution strategy by agent

| Agent | Solution Strategy Chosen | | N | $\chi^2(1)$ | p |
|---|---|---|---|---|---|
| | **Additive** | **Subtractive** | | | |
| Human | 335 (56.97%) | 253 (43.03%) | 1268 | 26.04 | <0.001 |
| GPT-4 | 481 (70.74%) | 199 (29.26%) | | | |

### Table 5 | Logistic regression analysis results for Experiments 1a and 1b

| Experiment | Task | Effect | N | $\chi^2(1)$ | p |
|---|---|---|---|---|---|
| Experiment 1a | Symmetry | sol eff | 302 | 1.67 | 0.196 |
| | | agent | | 6.02 | 0.014 |
| | | sol eff x agent | | 20.54 | <0.001 |
| Experiment 1b | Summary | sol eff | 341 | 0.32 | 0.574 |
| | | agent | | 50.30 | <0.001 |
| | | sol eff x agent | | 10.84 | <0.001 |

*Note.* sol eff = solution efficiency.

### Table 6 | Post hoc analysis results for Experiments 1a and 1b

| Experiment | Task | Agent | add and sub equally eff | | sub more eff | | $\chi^2(1)$ | p |
|---|---|---|---|---|---|---|---|---|
| | | | **N** | **add. strategy (in %)** | **N** | **add. strategy (in %)** | | |
| Experiment 1a | Symmetry | Human | 67 | 65.67 | 65 | 29.23 | 19.95 | <0.001 |
| | | GPT-4 | 85 | 54.12 | 85 | 69.41 | 4.27 | 0.039 |
| Experiment 1b | Summary | Human | 87 | 57.47 | 84 | 39.29 | 5.78 | 0.016 |
| | | GPT-4 | 85 | 77.65 | 85 | 90.59 | 5.44 | 0.020 |

*Note.* add and sub equally eff = addition and subtraction equally efficient; sub more eff = subtraction more efficient (than addition); add. strategy = additive solution strategy.

**Hypotheses 2 and 3: impact of solution efficiency and valence of instruction on choice of additive solution strategy.** Table 12 summarizes the descriptive frequencies of additive solution strategies by agent and experimental condition in Experiments 3a and 3b. Table 13 presents the corresponding logistic regression analyses. In Experiment 3a (symmetry task), significant main effects of solution efficiency and agent were observed, as well as a significant interaction between the two factors. No additional main effects, two-way interactions, or the three-way interaction between solution efficiency, valence of instruction, and agent reached significance. In Experiment 3b (summary task), a more complex picture emerged. All three main effects (solution efficiency, valence of instruction, and agent) were significant, as was the two-way interaction between valence of instruction and agent. Additionally, the three-way interaction between solution efficiency, valence of instruction, and agent reached significance.

Given the significant three-way interaction in Experiment 3b, the two-way interaction between solution efficiency and instruction valence was examined separately within each agent condition. For humans, the two-way interaction was not significant, $\chi^2(1) = 0.01$, $p = 0.912$, indicating independent effects of the two factors. For GPT-4o, the two-way interaction was significant, $\chi^2(1) = 4.20$, $p = 0.041$, indicating that the effect of solution efficiency depended on instruction valence: under neutral instructions, additive strategy selection dropped when subtraction was more efficient relative to equal efficiency (88.15% vs. 98.52%); under positive instructions, solution efficiency had no meaningful impact (100.00% vs. 99.26%), given that additive strategy selection was already at ceiling. To further disentangle these effects and evaluate Hypotheses 2 and 3, post hoc analyses were conducted for Experiments 3a and 3b.

Hypothesis 2: impact of solution efficiency on choice of additive solution strategy. The results of the post hoc analyses of solution efficiency in Experiments 3a and 3b are shown in Table 14. In Experiment 3a (symmetry task), post hoc analyses were warranted by the significant interaction between solution efficiency and agent. Human participants selected additive solution strategies significantly more often when addition and subtraction were equally efficient than when subtraction was more efficient, consistent with the hypothesis. In contrast, no significant difference was found for

GPT-4o, which is inconsistent with the hypothesis. In Experiment 3b (summary task), post hoc analyses were carried out, although the overall model revealed a significant three-way interaction, to explore the direction of the effects. In the summary task, for both humans and GPT-4o, additive strategies were produced more frequently when addition and subtraction were equally efficient than when subtraction was more efficient, consistent with the hypothesis.

Hypothesis 3: impact of valence of instruction on choice of additive solution strategy. Table 15 summarizes the results of the post hoc analyses on the valence of instruction in Experiments 3a and 3b. For Experiment 3a (symmetry task), post hoc analyses are reported solely for completeness. As neither main effects nor interactions involving valence reached significance, these results are not meaningful for inference. Accordingly, no significant differences were observed between neutral and positive instructions for either humans or GPT-4o, i.e., the data do not provide empirical support for the hypothesized valence effect in this task. In Experiment 3b (summary task), post hoc analyses were conducted based on the significant main effects and the significant two- and three-way interactions involving valence. In the summary task, under positive instructions, both humans and GPT-4o generated more additive solution strategies than under neutral instructions, consistent with the hypothesis.

**Additional analyses: accuracy of solutions.** As in Study 1, accuracy rates differed between humans and the GPT model, although the extent of this difference largely depended on the task and output format (see Table 16). Humans demonstrated consistently high accuracy (69–95%). In Experiment 3b (summary task), GPT-4o rarely produced correct solutions (0–9%). In Experiment 3a (symmetry task), GPT-4o's accuracy varied considerably depending on the output format: while accuracy in the text-based format was relatively low (7–15%), accuracy in the grid-based format was markedly higher (61–73%), enabling follow-up analyses limited to correct solutions for this task, with the grid-based format being filtered for correct answers.

Follow-up analyses focusing only on accurate solutions were carried out for Experiment 3a (symmetry task) to check if the previously observed patterns remain consistent when considering only accurate trials ($N = 609$, with $N = 255$ for human answers and $N = 354$ for GPT-4o grid-based outputs). Results indicated that the overall pattern was the same as in the analyses based on the full dataset. Across all accurate responses, additive solution strategies were chosen in 456 cases (74.88%) and subtractive strategies in 153 cases (25.12%), $\chi^2(1) = 150.75$, $p < 0.001$, $\Phi = 0.50$, 95% CI [0.42, 0.58]. The chi-square test of homogeneity indicated a significant difference in solution strategy distributions between agents, $\chi^2(1) = 185.59$, $p < 0.001$, $\Phi = 0.55$, 95% CI [0.47, 0.63]. Human participants ($N = 255$) selected additive solutions in 119 cases (46.67%) and subtractive solutions in 136 cases (53.33%), whereas GPT-4o ($N = 354$) selected additive solutions in 337 cases (95.20%) and subtractive solutions in 17 cases (4.80%).

The logistic regression revealed significant main effects of agent, $\chi^2(1) = 199.75$, $p < 0.001$, and solution efficiency, $\chi^2(1) = 35.47$, $p < 0.001$, as well as a significant interaction between agent and solution efficiency, $\chi^2(1) = 7.07$, $p = 0.008$. No main or interaction effects involving instruction

**Table 7 | Logistic regression analysis results for Experiments 2a and 2b**

| Experiment | Task | Effect | N | $\chi^2(1)$ | p |
|---|---|---|---|---|---|
| Experiment 2a | Symmetry | val ins | 289 | 0.62 | 0.430 |
| | | agent | | 3.85 | 0.050 |
| | | val ins x agent | | 0.95 | 0.329 |
| Experiment 2b | Summary | val ins | 336 | 10.59 | 0.001 |
| | | agent | | 15.20 | <0.001 |
| | | val ins x agent | | 6.84 | 0.009 |

*Note.* val ins = valence of instruction.

**Table 8 | Post hoc analysis results for Experiments 2a and 2b**

| Experiment | Task | Agent | val neutral | | val positive | | $\chi^2(1)$ | p |
|---|---|---|---|---|---|---|---|---|
| | | | N | add. strategy (in %) | N | add. strategy (in %) | | |
| Experiment 2a | Symmetry | Human | 60 | 63.33 | 59 | 61.02 | 0.07 | 0.796 |
| | | GPT-4 | 85 | 45.88 | 85 | 55.29 | 1.50 | 0.220 |
| Experiment 2b | Summary | Human | 83 | 65.06 | 83 | 73.49 | 1.38 | 0.240 |
| | | GPT-4 | 85 | 76.47 | 85 | 96.47 | 15.70 | <0.001 |

*Note.* val neutral = valence of instruction neutral; val positive = valence of instruction positive; add. strategy = additive solution strategy.

valence were significant (see Supplementary Information SI 7, Table A34, for a complete overview of results).

Post hoc analyses comparing solution efficiency indicated that humans chose additive strategies in 79 out of 115 cases (68.70%) when addition and subtraction were equally efficient, and in 40 out of 140 cases (28.57%) when subtraction was more efficient, $\chi^2(1) = 41.91$, $p < 0.001$. For GPT-4o, additive strategies occurred in 173 out of 181 cases (95.58%) under equal efficiency and in 164 out of 173 cases (94.80%) when subtraction was more efficient, $\chi^2(1) = 0.12$, $p = 0.731$.

Post hoc analyses based on instruction valence showed that humans used additive strategies in 60 out of 133 cases (45.11%) under neutral and 59 out of 122 cases (48.36%) under positive instructions, $\chi^2(1) = 0.27$, $p = 0.604$. For GPT-4o, additive strategies appeared in 172 out of 182 cases (94.51%) under neutral and 165 out of 172 cases (95.93%) under positive instructions, $\chi^2(1) = 0.40$, $p = 0.530$.

## Table 9 | Overview of accuracy of solutions proposed by humans and by GPT-4 (Experiments 1a, 1b, 2a, 2b)

| Experiment / Condition | Human samples | | GPT-4 samples | |
|---|---|---|---|---|
| | Accurate Solutions (*n*) | Inaccurate Solutions (*n*) | Accurate Solutions (*n*) | Inaccurate Solutions (*n*) |
| Experiment 1a | | | | |
| add and sub equally eff | 65 (97.01%) | 2 (2.99%) | 2 (2.35%) | 83 (97.65%) |
| sub more eff | 52 (80.00%) | 13 (20.00%) | 2 (2.35%) | 83 (97.65%) |
| Experiment 1b | | | | |
| add and sub equally eff | 74 (85.06%) | 13 (14.94%) | 15 (17.65%) | 70 (82.35%) |
| sub more eff | 73 (86.90%) | 11 (13.10%) | 13 (15.29%) | 72 (84.71%) |
| Experiment 2a | | | | |
| val neutral | 58 (96.67%) | 2 (3.33%) | 1 (1.18%) | 84 (98.82%) |
| val positive | 55 (93.22%) | 4 (6.78%) | 3 (3.53%) | 82 (96.47%) |
| Experiment 2b | | | | |
| val neutral | 68 (81.93%) | 15 (18.07%) | 15 (17.65%) | 70 (82.35%) |
| val positive | 72 (86.75%) | 11 (13.25%) | 57 (67.06%) | 28 (32.94%) |

*Note.* Solutions were considered as accurate if the proposed solution in the symmetry task resulted in a grid pattern that was perfectly symmetrical from left to right and from bottom to top (Experiments 1a and 2a) or if the word count of the edited text in the summary task was outside the prohibited ranges as specified (Experiments 1b and 2b). add and sub equally eff = addition and subtraction equally efficient; sub more eff = subtraction more efficient (than addition); val neutral = valence of instruction neutral; val positive = valence of instruction positive.

## Table 10 | Frequencies and chi-square results for chosen solution strategy

| Solution Strategy Chosen | | *N* | $\chi^2(1)$ | *p* |
|---|---|---|---|---|
| Additive | Subtractive | | | |
| 1446 (78.97%) | 385 (21.03%) | 1831 | 614.81 | <0.001 |

## Table 11 | Frequencies and chi-square results for chosen solution strategy by agent

| Agent | Solution Strategy Chosen | | *N* | $\chi^2(1)$ | *p* |
|---|---|---|---|---|---|
| | Additive | Subtractive | | | |
| Human | 407 (54.19%) | 344 (45.81%) | 1831 | 470.78 | <0.001 |
| GPT-4o | 1039 (96.20%) | 41 (3.80%) | | | |

## Discussion

Study 1 set out with three objectives: (1) to investigate human addition bias and replicate prior research with inconsistent findings, (2) to extend this line of research to the GPT-4 LLM, and (3) to explore differences between humans and GPT-4 in the selection of additive and subtractive strategies. To this end, efficiency and instruction valence were manipulated independently and systematically in both a spatial (symmetry) and a linguistic (summary) problem.

Overall, human participants showed a preference for additive strategies, consistent with the addition bias reported by Adams et al. [1] and Fillon et al. [3]. At the same time, participants were sensitive to efficiency manipulations: when subtraction was more efficient, the frequency of additive solutions declined. These results extend prior evidence by showing that addition bias is robust, but not invariant. In line with theoretical accounts that highlight heuristic accessibility[1], humans defaulted to additive solutions, yet integrated cost-benefit information when efficiency cues were salient.

GPT-4 also displayed a strong preference for additive strategies, confirming that addition bias generalizes to outputs of current LLMs. Unlike human participants, GPT-4 did not incorporate efficiency constraints. Instead, the bias was even amplified: additive strategies occurred more frequently when subtraction offered a more efficient path than when both addition and subtraction required similar effort. This suggests that the addition bias, inherently present in human language (see ref. [16]), is further amplified by GPT-4 (see ref. [48]), as the outputs of the LLM appear to be influenced more by a strong additive default than by an adaptive integration of solution efficiency. The valence manipulation points to the role of affective linguistic cues: GPT-4 produced more additive solutions under positively framed instructions than under neutrally framed instructions in the summary task. This might be in line with the argument that human linguistic associations – in this case the stronger relation between addition and verbs of improvement as compared to verbs of change[16] – are not only mirrored but even amplified by LLMs[48].

The findings demonstrate systematic differences between humans and LLMs regarding additive and subtractive strategy selection. Humans combined default additive preferences with efficiency-based adjustments; GPT-4 did not. Conversely, GPT-4 outputs reflected lexical-semantic framing effects, whereas no statistically reliable effect of these manipulations was observed for humans. These divergences suggest that addition bias in humans is constrained by cognitive control and cost-benefit integration,

## Table 12 | Percentages of additive solution strategy for Experiments 3a and 3b by agent and condition

| Experiment | Task | Agent | add and sub equally eff + val neutral | | sub more eff + val neutral | | add and sub equally eff + val positive | | sub more eff + val positive | |
|---|---|---|---|---|---|---|---|---|---|---|
| | | | *N* | add. strategy (in %) | *N* | add. strategy (in %) | *N* | add. strategy (in %) | *N* | add. strategy (in %) |
| Experiment 3a | Symmetry | Human | 66 | 71.21 | 95 | 37.89 | 57 | 66.67 | 91 | 42.86 |
| | | GPT-4o | 135 | 94.07 | 135 | 94.81 | 135 | 97.04 | 135 | 97.78 |
| Experiment 3b | Summary | Human | 102 | 54.90 | 107 | 42.06 | 114 | 69.30 | 119 | 56.30 |
| | | GPT-4o | 135 | 98.52 | 135 | 88.15 | 135 | 99.26 | 135 | 100.00 |

*Note.* add and sub equally eff = addition and subtraction equally efficient; sub more eff = subtraction more efficient (than addition); val neutral = valence of instruction neutral; val positive = valence of instruction positive; add. strategy = additive solution strategy.

whereas in GPT-4, it is primarily a byproduct of distributional language patterns.

It is important to note that these conclusions are subject to several limitations. First, Study 1 tested only one language model, which limits the generalizability of the findings to other architectures or versions. Second, GPT-4 produced low rates of accurate solutions in both tasks, raising concerns about the functional adequacy of its outputs even when strategy patterns could be coded. Third, the valence manipulation was relatively weak, as the goal verb ("change" / "edit" or "improve") appeared only once in the symmetry task and twice in the summary task, potentially reducing its salience. Finally, efficiency and valence were manipulated independently rather than in a fully crossed design, preventing systematic tests of their interaction. These limitations of Study 1 underscore the need for replication with improved manipulations and design features, which motivated the fully crossed and strengthened design of Study 2.

Study 2 was concerned with the realization of two primary objectives: (1) the replication and expansion of the findings of Study 1 by substituting GPT-4 with GPT-4o; and (2) the addressing of the methodological limitations of Study 1, e.g., through the implementation of a fully crossed design that manipulated solution efficiency and the valence of instruction simultaneously across both tasks.

Across conditions, both humans and GPT-4o again showed a preference for additive over subtractive solutions. This replicates the robustness of the addition bias across GPT models and study design. GPT-4o exhibited stronger overall addition bias than humans, even stronger than in Study 1. For the LLM, the choice of solution strategy showed a ceiling effect, with additive strategies being used at a frequency of 88% to 100% in the given conditions. This indicates that the LLM might be particularly susceptible to additive transformations when engaging in problem-solving tasks.

Study 2 replicated the findings that human participants adaptively integrated efficiency cues, choosing additive strategies more often when addition and subtraction were equally efficient, compared to when subtraction was more efficient. GPT-4o demonstrated a comparable deficit in adapting to efficiency constraints as GPT-4 in the symmetry task, while in the summary task, its output patterns exhibited a stronger alignment with those of the human sample. This pattern reinforces the conclusion that LLMs are less sensitive to structural efficiency constraints than humans, especially in spatial problems.

While in Study 1, the valence of instruction had no effect on humans and only a weak, task-specific effect on GPT-4, in Study 2, stronger valence effects emerged in the summary task: under positive instructions, both humans and GPT-4o showed higher proportions of additive transformations. This suggests that the strengthened manipulation (goal verbs repeated four times rather than once or twice) might have successfully enhanced salience, enabling valence cues to influence strategy selection. In contrast, the symmetry task continued to yield non-significant results for valence in both agents, thereby raising the question of whether affective framing exerts a negligible effect in this spatial domain or whether the present manipulation lacked sufficient salience.

Follow-up analyses focused on accurate solutions in the symmetry task confirmed that the pattern remained consistent. Both agents continued to exhibit a clear addition bias; this bias remained significantly stronger in GPT-4o than in human participants. Patterns related to solution efficiency and the valence of instruction were also comparable to those of all solutions. The persistence of these effects after excluding inaccurate trials indicates that the addition bias is not due to random responding or comprehension errors but reflects a stable tendency.

**Table 13 | Logistic regression analysis results for Experiments 3a and 3b**

| Experiment | Task | Effect | N | χ²(1) | p |
|---|---|---|---|---|---|
| Experiment 3a | Symmetry | sol eff | 849 | 17.66 | <0.001 |
| | | val ins | | 1.05 | 0.305 |
| | | agent | | 232.91 | <0.001 |
| | | sol eff x val ins | | 0.69 | 0.407 |
| | | sol eff x agent | | 8.07 | 0.004 |
| | | val ins x agent | | 2.43 | 0.119 |
| | | sol eff x val ins x agent | | 0.06 | 0.803 |
| Experiment 3b | Summary | sol eff | 982 | 13.97 | <0.001 |
| | | val ins | | 19.03 | <0.001 |
| | | agent | | 269.13 | <0.001 |
| | | sol eff x val ins | | 0.05 | 0.825 |
| | | sol eff x agent | | 3.46 | 0.063 |
| | | val ins x agent | | 9.05 | 0.003 |
| | | sol eff x val ins x agent | | 4.16 | 0.041 |

*Note.* sol eff = solution efficiency; val ins = valence of instruction.

**Table 14 | Post hoc analysis results for Experiments 3a and 3b by agent and solution efficiency**

| Experiment | Task | Agent | add and sub equally eff | | sub more eff | | χ²(1) | p |
|---|---|---|---|---|---|---|---|---|
| | | | N | add. strategy (in %) | N | add. strategy (in %) | | |
| Experiment 3a | Symmetry | Human | 123 | 69.11 | 186 | 40.32 | 25.04 | <0.001 |
| | | GPT-4o | 270 | 95.56 | 270 | 96.30 | 0.19 | 0.663 |
| Experiment 3b | Summary | Human | 216 | 62.50 | 226 | 49.56 | 7.53 | 0.006 |
| | | GPT-4o | 270 | 98.89 | 270 | 94.07 | 10.09 | 0.001 |

*Note.* add and sub equally eff = addition and subtraction equally efficient; sub more eff = subtraction more efficient (than addition); add. strategy = additive solution strategy.

**Table 15 | Post hoc analysis results for Experiments 3a and 3b by agent and valence of instruction**

| Experiment | Task | Agent | val neutral | | val positive | | χ²(1) | p |
|---|---|---|---|---|---|---|---|---|
| | | | N | add. strategy (in %) | N | add. strategy (in %) | | |
| Experiment 3a | Symmetry | Human | 161 | 51.55 | 148 | 52.03 | 0.01 | 0.934 |
| | | GPT-4o | 270 | 94.44 | 270 | 97.41 | 3.10 | 0.078 |
| Experiment 3b | Summary | Human | 209 | 48.33 | 233 | 62.66 | 9.21 | 0.002 |
| | | GPT-4o | 270 | 93.33 | 270 | 99.63 | 19.06 | <0.001 |

*Note.* val neutral = valence of instruction neutral; val positive = valence of instruction positive; add. strategy = additive solution strategy.

**Table 16 | Overview of accuracy of solutions proposed by humans and by GPT-4o (experiments 3a and 3b)**

| Experiment / Condition | Human samples | | GPT-4o samples (text-based response format) | | GPT-4o samples (grid-base response format) | |
|---|---|---|---|---|---|---|
| | Accurate Solutions (n) | Inaccurate Solutions (n) | Accurate Solutions (n) | Inaccurate Solutions (n) | Accurate Solutions (n) | Inaccurate Solutions (n) |
| **Experiment 3a** | | | | | | |
| add and sub equally eff + val neutral | 61 (92.42%) | 5 (7.58%) | 11 (8.15%) | 124 (91.85%) | 98 (72.59%) | 37 (27.41%) |
| sub more eff + val neutral | 72 (75.79%) | 23 (24.21%) | 19 (14.07%) | 116 (85.93%) | 84 (62.22%) | 51 (37.78%) |
| add and sub equally eff + val positive | 54 (94.74%) | 3 (5.26%) | 14 (10.37%) | 121 (89.63%) | 83 (61.48%) | 52 (38.52%) |
| sub more eff + val positive | 68 (74.73%) | 23 (25.27%) | 10 (7.41%) | 125 (92.59%) | 89 (65.93%) | 46 (34.07%) |
| **Experiment 3b** | | | | | | |
| add and sub equally eff + val neutral | 81 (79.41%) | 21 (20.59%) | 5 (3.70%) | 130 (96.30%) | | |
| sub more eff + val neutral | 79 (73.83%) | 28 (26.17%) | 3 (2.22%) | 132 (97.78%) | | |
| add and sub equally eff + val positive | 83 (72.81%) | 31 (27.19%) | 11 (8.15%) | 124 (91.85%) | | |
| sub more eff + val positive | 83 (69.75%) | 36 (30.25%) | 1 (0.74%) | 134 (99.26%) | | |

*Note.* Solutions were considered as accurate if the proposed solution in the symmetry task resulted in a grid pattern that was perfectly symmetrical from left to right and from bottom to top (Experiment 3a) or if the word count of the edited text in the summary task was outside the prohibited ranges as specified (Experiment 3b). add and sub equally eff = addition and subtraction equally efficient; sub more eff = subtraction more efficient (than addition); val neutral = valence of instruction neutral; val positive = valence of instruction positive.

Study 2 effectively addressed the main methodological limitations identified in Study 1 through a series of targeted design improvements. Most notably, the study employed a fully crossed $2 \times 2 \times 2$ design that simultaneously manipulated solution efficiency and instruction valence for both agents, enabling systematic testing of main and interaction effects that were not possible in Study 1. The valence manipulation was strengthened by repeatedly embedding the goal verb across the instructions, enhancing its linguistic and emotional salience. The instruction for the symmetry task was also redesigned to reduce interpretative ambiguity. Additionally, all responses were independently coded by two raters, ensuring greater reliability. A further methodological improvement was the coding of both text- and grid-based response formats for both humans and GPT-4o. Study 2 not only replicated the core outcomes of Study 1 with the successor model (GPT-4o) but also demonstrated that the observed patterns persist when potential accuracy-related artifacts are removed. This offers a more reliable empirical foundation for the subsequent General Discussion, which synthesizes results from both studies and explores their broader implications for human and artificial problem-solving.

### Theoretical integration

Humans tend to prefer additive transformations when modifying ideas, objects, and situations, while subtractive transformations are often overlooked, even though they may represent superior solutions. This addition bias was systematically investigated by Adams et al. [1], but – as the opening quote may imply – it probably existed for centuries, with implications for today's everyday life (see, e.g., refs. [17–19]).

The present research examined the prevalence and drivers of the addition bias in humans and LLMs, using GPT-4 (Study 1) and GPT-4o (Study 2). Across two studies and two task domains (spatial symmetry, linguistic summary), we manipulated both cognitive constraints (solution efficiency) and affective cues (instruction valence). Table [17] provides an overview of the results, indicating which hypotheses were supported (denoted by a check mark), which were non-significant (n.s.), and which were significant but in the opposite direction of the predicted effect, that is, contrary to the hypothesis (sig., contrary to hyp.) across both studies and tasks. Three central findings emerged.

First, the overall addition bias proved robust. Both humans and LLMs systematically favored additive over subtractive solution strategies. This replicates and extends earlier work on subtraction neglect[1,3], demonstrating that this bias generalizes to artificial agents trained on large-scale language data. This is consistent with evidence by Santagata and De Nobili[49] showing a "more-is-more" tendency in LLMs across tasks such as constructing palindromes, balancing LEGO towers, and summarizing texts, consistent with findings by Adams et al. [1] and partially observed in Studies 1 and 2. The direct comparison of overall addition bias between human participants and GPT-4 and GPT-4o in our studies further demonstrated that the tendency of favoring additive over subtractive solution strategies was even stronger and amplified in GPT-4 and GPT-4o compared to human participants.

Second, humans and LLMs differed systematically in their sensitivity to efficiency (as a cognitive driver of the bias). Humans consistently modulated their strategy use according to cost-benefit cues: additive defaults were attenuated when subtraction offered a more efficient path to the goal. In contrast, GPT-4 exhibited the opposite outcome pattern, generating even more additive solutions when subtraction would have been more efficient. While GPT-4o exhibited sensitivity to efficiency cues in neutral instructions, reducing additive responses when subtraction was more efficient, this sensitivity was absent under positively valenced instructions, where additive responding reached ceiling levels. This pattern suggests that the modulation of strategy use in the GPT models was contingent on contextual framing as opposed to genuine cost-benefit evaluation. In contrast to humans, whose strategy choices adapt flexibly to task efficiency, the GPT models' performances indicate an absence of strategy adaptation in response to task efficiency, and an overreliance on default additive transformations, especially when task framing was positively valenced.

**Table 17 | Overview of hypothesis testing results**

| Study | Task type | Both agents | Human | | GPT-4 (Study 1) / GPT-4o (Study 2) | |
|---|---|---|---|---|---|---|
| | | Overall AB | AB: eff equal > sub more eff | AB: val pos > val neutral | AB: eff equal > sub more eff | AB: val pos > val neutral |
| Study 1 | Spatial | ✓ | ✓ | n.s. | sig., contrary to hyp. | n.s. |
| | Lingustic | | ✓ | n.s. | sig., contrary to hyp. | ✓ |
| Study 2 | Spatial | ✓ | ✓ | n.s. | n.s. | n.s. |
| | Lingustic | | ✓ | ✓ | ✓ | ✓ |

*Note.* AB = addition bias; eff equal = addition and subtraction equally efficient; sub more eff = subtraction more efficient (than addition); val neutral = valence of instruction neutral; val positive = valence of instruction positive.

Third, the effects of valence (as an affective driver of the bias) were inconsistent. In Study 1, valence influenced GPT-4 only in the linguistic task, leaving human behavior unchanged. In Study 2, a stronger manipulation (goal verbs repeated four times) yielded significant valence effects in the linguistic task for both humans and GPT-4o: additive strategies were produced more frequently under positive than neutral instructions. This indicates that affective framing can bias problem-solving preferences, but its impact depends on the task domain and the salience of linguistic cues. The persistence of non-significant effects in the symmetry task is consistent with the idea that valence effects, if present, are weaker than structural constraints in this domain.

Whereas our primary focus was the choice of additive and subtractive solution strategies, we also analyzed solution accuracy. Unlike human participants who demonstrated decent problem-solving abilities, the overall accuracy rates for GPT models were low. The text-based solutions from both GPT-4 and GPT-4o appeared to lack an understanding of symmetry, as evidenced by the low rate of accurate solutions, supporting earlier research on LLMs' spatial abilities (see, e.g., ref. 37). Furthermore, both models consistently failed to produce correct solutions in the summary tasks. One explanation for this failure is the difficulty LLMs have in understanding negations[50], which was necessary to solve the summary tasks accurately (e.g., "The improved summary MUST NOT be between 64 and 88 words long."). These low problem-solving abilities stand in contrast to earlier research results (e.g., refs. 31,33,34). Given these low solution accuracies, one might question the meaningfulness of analyzing a problem-solving bias. Notably, the grid-based output of GPT-4o in Experiment 3a achieved significantly higher accuracy. The subsequent analysis, based on this subset of accurate solutions, showed that the primary pattern regarding solution strategy remained consistent with the findings from the low-accuracy, text-based output. This suggests that the analysis of the solution strategy yields meaningful patterns of results and is not an artifact of general response quality or comprehension failures.

Human users increasingly rely on LLM suggestions, and model outputs can influence downstream human reasoning and decision-making. This emphasizes the importance of systematically examining how and under which conditions additive tendencies emerge and are amplified in human-LLM interaction. At the same time, additive transformations should not be interpreted as inherently maladaptive. Future research should therefore investigate more systematically under which task demands, goal structures, and evaluative criteria additive versus subtractive transformations are functionally adaptive, and how such contextual dependencies differ between humans and LLMs.

## Limitations

This research has significant strengths and limitations. One strength lies in its conceptual framework and methodological approach, contributing to the growth of current knowledge about additive and subtractive solution strategies in both humans and LLMs. The topic is highly relevant both theoretically and practically. This is because systematic research on addition bias has only recently begun (see ref. 1) and is still marked by mixed results. To address this, a comparison framework was created where identical task structures were shown to human participants and GPT models. This approach allowed for the identification of differences in problem-solving behavior while reducing confounds related to task design, framing, or instruction wording. Using two different task domains enabled the exploration of whether the addition bias is consistent across spatial and linguistic problem-solving contexts. A notable strength of the study is the systematic adaptation of the original materials from Adams et al. [1]. These modifications ensured that the GPT models could not depend on memorized examples or task-specific phrasing from their potential training data, thereby reducing contamination from prior exposure. By standardizing task instructions across agents, the study enables direct comparisons between human and artificial output under consistent linguistic input conditions.

Despite the advantages mentioned, it is crucial to consider the methodological limitations when analyzing the findings. Presenting humans with linguistically detailed, non-interactive tasks limits ecological validity. Humans are generally unaccustomed to solving spatial problems purely verbally, as required in the symmetry task. Many participants merely repeated the task instruction in their own words or theoretically described how they would proceed without describing an actual solution. Difficulty in task comprehension may have led to a higher cognitive load, which, in itself, has been shown to influence addition bias[1]. During data analysis, it became clear that some participants in Study 1 misunderstood the instruction "switch fields on and off" as "switch fields around", resulting in unchanged grid states and subsequent data exclusion. This led to refinements in Study 2, where the verb "toggle" was used along with more precise guidance. However, relatively high exclusion rates, especially in the experiments involving the spatial task, indicate potential selection bias and a less representative human sample. Previous studies (see refs. 1,4) used interactive versions of the spatial task, allowing participants to solve problems through direct interaction rather than verbal description. Future research should adopt similarly interactive formats to promote more natural problem-solving and improve both ecological validity and task comprehension. Including brief comprehension checks, independent of the task, could also be beneficial to improve data quality in future research.

Both the symmetry and summary tasks should be regarded as simplified models of real-world problem-solving. Previous research, such as Adams et al. [1], employed tasks with similarly high levels of abstraction, e.g., participants modified minigolf courses or LEGO towers. Our study aimed to provide foundational insights into the addition bias rather than imitate realistic decision-making scenarios in detail. However, the overall naturalness of the tasks remains restricted. Verbal descriptions of spatial actions are seldom observed outside laboratory settings, and numerical word-count limits are artificial when considered in isolation. However, they reflect realistic constraints in professional communication, such as academic abstracts or application writing. Consequently, this work is a controlled study of basic tendencies rather than an ecological simulation of real-world reasoning. Future research could expand this approach by employing more naturalistic formats (e.g., revising slide layouts or concise rewrites of motivation letters) to explore the generalizability of addition bias across various contexts.

Some of the methodological critiques that Fischer et al. [51] raised towards the original paradigm by Adams et al. [1] also apply here. First, the order of cues for addition and subtraction was not counterbalanced. In contrast to cues presented by Adams et al., the implicit cue for subtraction was given first in task instructions (e.g., "switching fields from '[X]' to '[ ]' or from '[ ]' to '[X]'"), making subtraction more salient. However, no further conclusions can be drawn regarding a possible sequence effect. A second point of criticism is that the baseline numbers of filled and empty fields in the symmetry task were not equal. This design biases against subtraction and makes addition more likely. In other words, the influence of an inherent anchoring bias (see ref. [35]) might lead to a higher likelihood of adding filled fields when the baseline number of filled fields is smaller. Third, the patterns of filled fields presented in the symmetry task were not controlled for widespread cognitive spatial associations (addition is associated with right/up, subtraction with left/down)[52]. This might have increased the likelihood of additive solutions. Finally, the tasks used were similarly complex as the original tasks. According to Fischer et al. [51], this task complexity mixes processing domains, which can lead to confounding by cross-domain associations.

Conceptually, interpreting the similarities between human and LLM responses requires caution. While human addition bias likely reflects heuristic cognitive shortcuts and limitations in search or evaluation[1], corresponding patterns in the output of LLMs are more plausibly explained by the "entry points" of human regularities that enter these models through biased training data, annotation practices, and model design[23]. This distinction highlights that output similarity does not imply process similarity. A key methodological boundary of our work is that we analyzed outputs rather than processes. While this makes humans and LLMs directly comparable, it limits claims about internal mechanisms. As Shiffrin and Mitchell[53] emphasized, comparative studies must specify which aspect of LLMs – the verbal responses, the token probabilities, or the neural networks' internal representations – is under examination. To explore LLM's internal representations, future research should incorporate suitable methods such as chain-of-thought prompting. Similar to the unknown mechanisms of GPT-4o, our results do not further identify human mechanisms relevant to the bias, but they do document consistent response patterns under controlled manipulations. Our study followed cognitive-psychological conventions by focusing on verbal outputs. However, we acknowledge that more profound insights into the addition bias in both humans and LLMs would require alternative analytical approaches.

## Constraints on generality

The present studies entails constraints on generality regarding both humans and GPT models. Participants from both Study 1 and Study 2 were recruited exclusively from the United States, which limits the applicability of the results to other cultural contexts, as previous research has shown that addition bias varies across cultures[4]. This homogeneity, however, was deliberate: the cultural background was kept constant across studies to ensure comparability and enable a controlled replication of the effects observed in Study 1. This approach allowed attributing differences between Study 1 and 2 mainly to the experimental manipulations, solution efficiency and instruction valence, and the change in the language model rather than cultural variation.

Due to LLM's context dependence, which implies that small linguistic changes in prompts could lead to entirely different outcomes[9,53], generalizing the results is challenging. Instead of the deterministic approach, where most GPT studies typically choose a temperature parameter of 0, the parameter was kept at the default values of 0.7 (for GPT-4) and 1.0 (for GPT-4o) to achieve variability in the answers. This way, a "sample" of GPT trials could be collected. However, the parameter represents a black box, and the assumption that the variability resembles human variability is highly contestable. It remains unclear how to correctly interpret the results and whether they tell us anything about the "inner processes" of LLMs at all. Finally, as Shiffrin and Mitchell[53] remark, "psychological assessments designed to test humans' higher-level cognitive abilities […] may in fact not test these

abilities at all in LLMs, even when LLMs – trained on huge swaths of human-generated text – produce similar responses as humans" (p. 2). In other words, interpreting results obtained from LLMs in human terms is questionable and, in the long term, may become less and less feasible as AI models emerge[54]. Any generalizations must be made with caution.

Notably, we included GPT-4o– a successor version of GPT-4 – in Study 2 to assess whether the observed additive bias generalizes across model variants. The comparison between GPT-4 and GPT-4o revealed both replication and divergence. While GPT-4o reproduced the general tendency towards additive over subtractive solutions, indicating that this preference is not confined to a single model generation, e.g., the underlying sensitivity to solution efficiency differed. Future research should extend such comparisons to fundamentally different architectures (e.g., Claude, Gemini, LLaMA). Evidence indicates that the reflection of human biases may vary considerably between model families. For example, Strachan et al. [55] demonstrated notable differences between GPT and LLaMA2 models in how they exhibit and operationalize theory-of-mind reasoning. Nonetheless, our findings indicate a certain robustness of this bias across different GPT models.

## Considerations for future research

The findings revealed human addition bias in certain settings and a more pronounced addition bias in GPT models as compared to humans. Future research could incorporate less complex tasks as proposed by Fischer et al. [51] to explore differences between the agents on a more granular level. To gain deeper insights into the inner processes of humans and LLMs, more profound cognitive psychological methods may be useful, like the Cognitive Reflection Test[56]. For LLMs, future research may employ chain-of-thought prompting, which has been shown to enhance LLM reasoning abilities (see, e.g., refs. [26,57]), potentially making the problem-solving process more transparent and thereby more comparable. Such approaches could help disentangle the "entry points" of the addition bias for LLMs (see ref. [23]), thereby offering more transparent and interpretable comparisons between human and LLM reasoning. Future research should conduct systematic sensitivity analyses across multiple temperature settings, as default values such as 0.7 (for GPT-4) or 1.0 (for GPT-4o) lack empirical validation, to assess the impact of temperature variation on addition bias. For humans, think-aloud protocols[58] provide a better understanding of the task comprehension, solution generation, and solution choice processes, and hence, a better idea of where along the way addition bias occurs. The use of such methods might also shed light on which factors other than language influence human behavior and, hence, differentiate humans from LLMs. Within the NegativePrompt framework, Wang et al. [59] demonstrated that negative emotional stimuli can positively influence LLM's performance. This highlights the importance of understanding not only cognitive but also emotional factors in the interaction of humans and LLMs. Integrating frameworks like NegativePrompt[59] into the investigation might not only enrich our understanding of LLM biases but also emphasize the potential for emotional engagement strategies to enhance LLM performance across various applications. Building on this, future research should aim to strengthen valence manipulations further to better capture affective influences on problem-solving. This could include the use of richer contextual framing (e.g., embedding tasks in emotionally meaningful scenarios), explicit emotion induction (e.g., short affective priming before the task), or pre-tested valence manipulations to ensure they are perceived as intended. Emotional prompts can systematically enhance both humans' and LLMs' performance across deterministic and generative tasks, suggesting that affective framing may not only influence human reasoning but can also modulate model outputs[60]. Such improvements would enable more accurate testing of how emotional connotations interact with cognitive drivers, such as efficiency, in influencing the display of additive and subtractive strategies in both humans and LLMs.

Although we aimed to minimize ambiguity through refinements, we cannot dismiss the possibility that comprehension issues, rather than

biases, contributed to the effects observed in the human data. This confound should be addressed in future research by implementing measures such as interactive task formats. A related concern pertains to the LLM's dependence on patterns in its training data, which may influence its output tendencies in such tasks more than genuine problem-solving abilities. This raises the broader question of whether the model's responses reflect human cognitive biases, such as addition bias, or simply replicate patterns seen in the training corpus. Future studies should explore whether adjustments to training data or model architecture could reduce these biases and help differentiate between behaviors driven by training data regularities and those stemming from the model's inherent problem-solving mechanisms. As outlined by Ntoutsi et al.[48], bias mitigation in data-driven AI systems can occur at multiple stages, through data preprocessing (balancing or reweighting training samples), in-processing (integrating fairness constraints or regularization into model training), and post-processing (adjusting model outputs), each targeting distinct sources of bias within the learning pipeline.

Our findings raise important questions for the development of AI. To reduce biases in problem-solving, such as the observed addition bias – whether termed a bias or seen as a reflection of patterns – future research could focus on fine-tuning LLMs. For instance, using datasets that highlight effective subtractive strategies or minimal interventions, or employing contrastive learning approaches that favor parsimonious over excessive changes (e.g., methods like Direct Preference Optimization[61]) to align LLMs with human preferences. We present these as hypotheses for future technical research, not as definitive solutions. Systematic studies are necessary to evaluate the effectiveness of these adjustments.

Juvrud et al.[4] demonstrated that children aged 9-10 years use subtractive transformations less often than adults. This raises important questions about the developmental path and origins of addition bias. One possibility is that younger children's stronger preference for additive transformations reflects their relatively limited cognitive capacity, given that subtraction requires mentally representing the existing structure and then selectively removing elements[1], which is more demanding. In this view, age-related developments in executive functions – such as inhibitory control, working memory, and cognitive efficiency – may mediate the increased use of subtractive strategies. Alternatively, the bias may not decline but instead become more context-dependent with age, demonstrating increasing strategic flexibility or learned task framing. Future research should therefore investigate whether the addition bias weakens, stabilizes, or changes throughout development, and to what extent improvements in executive control influence those shifts.

Finally, other cognitive biases, such as the sunk cost fallacy or waste aversion, were mentioned as possible underlying factors of addition bias (see ref. 11). This study has not provided further insights into whether and how the addition bias differs from these concepts. Therefore, future research could control for other cognitive biases to delineate addition bias more clearly.

## Conclusion

To conclude, this research provides further evidence of human preference for additive solutions and reveals an even stronger addition bias in GPT-4 and GPT-4o. Across efficiency conditions, humans flexibly adapted their strategies – producing fewer additive responses when subtraction was more efficient than when addition and subtraction were equally efficient. In contrast, GPT-4 showed the opposite pattern, favoring additive strategies when subtraction provided a more efficient route. While GPT-4o was not significantly influenced by efficiency in the symmetry task, it did flexibly adapt its strategy (similar to the human pattern) in the summary task, where the pattern may have reversed (compared to GPT-4) due to improved task formulation. Moreover, statistically reliable valence effects were observed only in the linguistic domain, not in the spatial domain: humans remained unaffected by positive framing in Study 1, but not in Study 2, possibly due to higher affective and linguistic salience of the goal verbs. GPT models displayed a stronger addition bias when prompted to "improve" compared to

verbs of change in both Studies 1 and 2. These findings suggest that addition bias has been linguistically transferred to the LLMs, which has not only adopted but also amplified this preference for addition.

## Data availability
The data are available at https://osf.io/c78rm/.

## Code availability
The analysis scripts are available at https://osf.io/c78rm/.

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

## Acknowledgements
We thank Gerrit Anders for supporting us with collecting the GPT-4 and GPT-4o iterations. We thank Kevin Erens and Silas Seutemann for their support in coding responses for the symmetry task in Study 2.

## Author contributions
L.U. and V.J. conceptualized the study. L.U., V.J., J.B., M.H., and F.P. provided resources. L.U. and V.J. curated data and developed software. L.U. and V.J. performed formal analysis and created visualizations. L.U. and F.P. conducted the investigation. L.U., V.J., J.B., M.H., and F.P. contributed to methodology. L.U., V.J., and F.P. administered the project. V.J. wrote the original draft of the manuscript. L.U., V.J., J.B., M.H., and F.P. reviewed and edited the manuscript.

## Funding

## Competing interest

The authors declare no competing interests.
