## [Peer Review file · Communications Psychology]

Influence of Solution Efficiency and Valence of Instruction on Additive and Subtractive Solution Strategies in Humans, GPT-4, and GPT-4o

Corresponding Author: Ms Lydia Uhler

Version 0:

Decision Letter:

Dear Ms Uhler,

Thank you for your patience during the peer-review process. Your manuscript titled "Influence of Solution Efficiency and Valence of Instruction on Additive and Subtractive Solution Strategies in Humans and GPT-4" has now been seen by 3 reviewers, whose comments are appended below. You will see that they find your work of some potential interest. However, they have raised quite substantial concerns that must be addressed. In light of these comments, we cannot accept the manuscript for publication, but would be interested in considering a revised version that fully addresses these serious concerns.

We hope you will find the Reviewers' comments useful as you decide how to proceed. Should additional work allow you to address these criticisms, we would be happy to look at a substantially revised manuscript. If you choose to take up this option, please highlight all changes in the manuscript text file, and provide a detailed point-by-point reply to the reviewers.

Editorially, we consider it crucial that the empirical evidence supporting the central claim of the study is sufficiently strong and representative. To this end, please include an additional experiment with which the reviewers' concerns regarding operationalization of valence, task difficulty, and design (fully-crossed) are addressed. If possible, please also include a more diverse sample that includes participants from different cultural backgrounds. We strongly recommend avoiding any anthropomorphizing terms in relation to generative AI and LLMs.

I am attaching a checklist that details critical reporting requirements for the revised manuscript. Please attend to each item and ensure your manuscript is fully compliant. We are requesting that your manuscript aligns with these requirements as this facilitates the evaluation of your manuscript, reducing delays in re-review and potential future acceptance. If your revised manuscript is not aligned with these requests on major issues, such as those concerning statistics, it may be returned to you for further revisions without re-review. Additional information can be found in our style and formatting guide Communications Psychology formatting guide.

If the revision process takes significantly longer than five months, we will be happy to reconsider your paper at a later date, provided it still presents a significant contribution to the literature at that stage.

Please use the following link to submit your

- revised manuscript,
- point-by-point response to the referees' comments,
- cover letter (as a separate document),
- the Editorial Policy Checklist (see below),
- the Reporting Summary (see below), and
- the completed Editorial Request Table (attached):

Link Redacted

Thank you for the opportunity to review your work.

Best regards,

Troy Lui

Troy Lui, PhD
Associate Editor
Communications Psychology

REVIEWER EXPERTISE:

Reviewer #1: problem solving/ solution efficiency/ decision making, artificial intelligence

Reviewer #2: problem solving/ solution efficiency/ decision making

Reviewer #3: problem solving/ solution efficiency/ decision making, artificial intelligence

REVIEWER REPORTS:

Reviewer #1 (Remarks to the Author):

1. Summary

This paper aims to explore how the efficiency of solutions and the valence of instructions influence the choice between additive or subtractive strategies when humans and generative artificial intelligence (particularly GPT-4) solve problems. Through four pre-registered experiments, the study investigates 588 American participants and 680 iterations of GPT-4 across spatial and language tasks. The research finds that GPT-4 exhibits a stronger additive bias than humans under most conditions, showing a greater tendency to choose strategies that increase elements. Human participants are less inclined to select additive strategies when subtractive strategies are more efficient, whereas GPT-4 displays the opposite trend under the same conditions. Additionally, the valence of instructions significantly affects GPT-4's strategy choices but has no significant impact on humans. The study concludes that cognitive biases in GPT-4 are amplified in certain aspects, suggesting caution when utilizing such models in practical applications.

2. Strengths

Novel and Practically Relevant Research Topic: This paper focuses on the additive bias in problem-solving between humans and LLMs, filling a gap in existing research regarding cognitive biases in LLMs. It holds high academic value and practical significance.

Rigorous Methodological Design: The study employs four pre-registered experiments covering both spatial and language tasks, controlling for solution efficiency and the valence of instructions. The experimental design is comprehensive and replicable.

Adequate Sample Size: The study meets the expected sample sizes for both human participants and GPT-4 iterations, enhancing the statistical power and credibility of the results.

3. Weaknesses

Task Comprehension Difficulties Affect Data Quality: Some human participants struggled to understand the instructions in spatial tasks, resulting in the exclusion of a significant amount of data. This not only impacted the final sample size but may also introduce selection bias, affecting the representativeness of the results.

Insufficient Accuracy in GPT-4's Task Solutions: GPT-4 showed lower accuracy in solving spatial and summarization tasks, especially those requiring the understanding of complex instructions and negations. This calls into question the significance of analyses based on solution biases, as the model's inability to accurately complete tasks may complicate the interpretation of biased results.

Inadequate Operationalization of Instruction Valence: Although the study attempted to manipulate the valence of instructions by changing verbs (e.g., "change" vs. "improve"), the results showed no significant impact on humans. This may be due to

the manipulation of valence not being strong or specific enough to effectively elicit the intended emotional responses.

Limited Cultural Diversity Restricts Generalizability: Participants were exclusively from the United States, ignoring how different cultural backgrounds might influence additive biases. This limits the applicability and generalizability of the findings on a global scale.

Lack of Exploration into LLMs' Internal Processes: While the study suggests that GPT-4 may possess cognitive mechanisms favoring addition, it lacks an in-depth analysis of the model's internal workings, such as the application of chain-of-thought prompting methods. This omission prevents a detailed understanding of the specific reasons behind the formation of biases in LLMs.

Reviewer #2 (Remarks to the Author):

Review of
"Influence of Solution Efficiency and Valence of Instruction on Additive and Subtractive Solution: Strategies in Humans and GPT-4" by Uhler et al.

OVERALL ASSESSMENT

The study examines additive and subtractive strategies in humans and GPT-4 in problem-solving tasks, taking into account solution efficiency and instruction valence. Both humans and GPT-4 showed the presence of addition bias (AB). Importantly, GPT-4 had a stronger additive bias, especially in the positive valence condition. Also, GPT-4's decision-making process (DM) was less flexible than that of human participants, whose decisions were overall more efficient with regard to the outcome and less prone to the addition bias alone.

The study is quite timely and relevant as it informs our understanding of the mechanisms underlying AI performance as a potential aid to human DM. Importantly, the paper extends previously reported data on the addition bias in LLMs by using both spatial and linguistic tasks and examining the contribution of cognitive and affective factors (i.e., solution efficiency and instruction valence), which expands the report's theoretical value. The paper is well-written, the design is very elegant, and the statistical analyses are appropriate. I recommend minor revisions based on my comments provided below.

COMMENTS

Most of my comments focus on important limitations that were not mentioned or that were not discussed in enough depth. However, one major comment has to do with the overall theoretical importance of the paper. I admit this may necessitate a substantial degree of rewriting, so I leave it to the editor to decide if this is necessary before the paper can be accepted for publication. Nevertheless, my view is that any paper comparing AI performance to that of humans needs to explain how the reported data potentially answer one of the following general research questions, so I would prefer a deeper discussion of these:

- A. How accurately/to what degree does the training corpus used by different LLMs reflect human cognitive biases (and cognitive processes in general)? For example, some biases may have stronger sources in discourse rather than written texts. Similarly, the training corpora may have subsections that are suited better/worse to answer this question.
- B. What does analysis of LLM's performance tell us about human cognition?
- C. What does analysis of LLM's performance tell us about LLM's "cognition"?
- D. How can LLM algorithms, etc., be improved in the future?

OTHER COMMENTS

1. p. 3. The para starting with "Both..." needs a bit more work as it leaves the reader with little understanding of what it may mean that 9–10-year-olds use subtractive transformations less than adults. May it mean that what children are lacking is efficiency, which is sometimes achieved via subtraction instead of addition? May it also mean that addition bias is not enhanced by experience? Can it be that it becomes weaker with age instead, and what would that mean?

2. Theoretical motivation of the task selection:

Apart from AB, the inclusion of H2 and H3 on p. 6 is weakly motivated. Please expand Intro to offer stronger motivation to include these parameters.

3. Experimental tasks and the comparability between LLM's and human DM or problem solving:

GPT-4's failure to produce accurate solutions in spatial tasks may reflect limitations in understanding abstract concepts rather than a general property of DM both in humans and LLMs. In other words, the DM process may be intrinsically different in AI and in humans, and the same experimental task may be capturing somewhat different "cognitive" processes. In other words, while authors use the terms decision-making and problem solving when talking about human and AI performance, these may well be very different "cognitive" processes.

4. Linguistic task:

The symmetry task conditions have different levels of linguistic complexity, which may have led to comprehension problems

rather than a modulation of AB in human participants. This is a potential confound that should be acknowledged as a limitation and a potential future direction. Same is true re. GPT-4's reliance on training corpus rather than problem-solving tendencies per se. This leads to another question: How accurately does the training corpus represent human AB? Future studies could assess whether modifications in training data or algorithmic adjustments reduce these biases.

5. Cultural constraints:

The human sample was limited to U.S.-based participants, restricting the study's cultural generalizability. This is especially important in light of the findings reported in Juvrud et al. (2024) showing substantial cultural differences in the attribution of AB.

6. Implications for AI Development:

It is not particularly clear to me what algorithm or training corpus adjustments would make LLMs more optimal/efficient in terms of AB.

7. Using other LLMs:

Some recent studies showed important differences in how individual LLMs represent/reflect human biases (e.g., Strachan, J. W., Albergo, D., Borghini, G., Pansardi, O., Scaliti, E., Gupta, S.,... & Becchio, C. (2024). Testing theory of mind in large language models and humans. *Nature Human Behaviour*, 1-11.) It may well be that there are differences in how AB is represented in different LLMs as well. This would improve the AI generalizability of the reported data and offer a more nuanced understanding of AI's problem-solving strategies.

Reviewer #3 (Remarks to the Author):

Please see the attached PDF file.

EDITORIAL POLICIES

We ask that you ensure your manuscript complies with our editorial policies and reporting requirements.

To that end, we require revised manuscripts to be accompanied by two completed items: a reporting summary that collects information on study design and procedure, and an editorial policy checklist that verifies compliance with all required editorial policies

- <https://www.nature.com/documents/nr-reporting-summary.zip>>Nature Research Reporting Summary
- <https://www.nature.com/documents/nr-editorial-policy-checklist.pdf>>Editorial Policy Checklist

All points on the policy checklist must be addressed. Your revised manuscript can only be sent back to the referees if these checklists are completed and uploaded with the revision.

Notes: If you have submitted a Stage 1 Registered Report, Review, Primer, Comment, or Perspective you do not need to submit these forms. If you have already submitted these forms, you may disregard this request.

** Visit Nature Research's author and referees' website at <http://www.nature.com/authors>>www.nature.com/authors for information about policies, services and author benefits**

Communications Psychology is committed to improving transparency in authorship. As part of our efforts in this direction, we are now requesting that all authors identified as 'corresponding author' create and link their Open Researcher and Contributor Identifier (ORCID) with their account on the Manuscript Tracking System prior to acceptance. ORCID helps the scientific community achieve unambiguous attribution of all scholarly contributions. You can create and link your ORCID from the home page of the Manuscript Tracking System by clicking on 'Modify my Springer Nature account' and following the instructions in the link below. Please also inform all co-authors that they can add their ORCIDs to their accounts and that

they must do so prior to acceptance.

Version 1:

Decision Letter:

Dear Ms Uhler,

Your manuscript titled "Influence of Solution Efficiency and Valence of Instruction on Additive and Subtractive Solution Strategies in Humans, GPT-4, and GPT-4o" has now been seen by our reviewers, whose comments appear below. In light of their advice I am delighted to say that we are happy, in principle, to publish a suitably revised version in Communications Psychology.

We therefore invite you to revise your paper one last time to address the remaining concerns of our reviewers and a list of editorial requests. At the same time we ask that you edit your manuscript to comply with our format requirements and to maximise the accessibility and therefore the impact of your work.

EDITORIAL REQUESTS:

SUBMISSION INFORMATION:

OPEN ACCESS:

*** TRANSPARENT PEER REVIEW:** Communications Psychology uses a transparent peer review system. On author request, confidential information and data can be removed from the published reviewer reports and rebuttal letters prior to publication. If you are concerned about the release of confidential data, please let us know specifically what information you would like to have removed. Please note that we cannot incorporate redactions for any other reasons.

*** CODE AVAILABILITY:** All Communications Psychology manuscripts must include a section titled "Code Availability" at the end of the methods section. We require that the custom analysis code supporting your conclusions is made available in a publicly accessible repository at this stage; please choose a repository that generates a digital object identifier (DOI) for the code; the link to the repository and the DOI must be included in the Code Availability statement. Publication as Supplementary Information will not suffice.

*** DATA AVAILABILITY:**

Link Redacted

Best regards,

Troy Lui

Troy Lui, PhD
Associate Editor
Communications Psychology

REVIEWERS' COMMENTS:

Reviewer #2 (Remarks to the Author):

OVERALL ASSESSMENT

Thank you for your comprehensive responses to my comments and for conducting Study 2, which significantly strengthens the manuscript. I appreciate the substantial revisions made throughout the paper, particularly the improved theoretical framing and discussion of limitations. Below are my comments on the revised manuscript.

REVISION 1 COMMENTS

Comment 2.1:

I am satisfied with how you have addressed the theoretical positioning of your work, particularly regarding points A-D. The clarifications about what your comparisons can and cannot reveal are much clearer now. The acknowledgment that similarities in output patterns do not imply shared mechanisms is appropriate and prevents overinterpretation of the findings. The discussion of bias transmission pathways (Osborne et al., 2023) provides valuable context. However, I would encourage slightly more elaboration on point (A) regarding how the training corpus might differentially represent additive vs. subtractive strategies in text. For example, are there linguistic or discourse features that might make additive transformations more frequent or salient in written corpora? This could provide additional theoretical depth to explain why GPT models show stronger additive bias than humans.

Comment 2.2:

The relocation and expansion of the discussion regarding Juvrud et al.'s (2024) developmental findings is an improvement. The acknowledgment of multiple possible interpretations (bias weakening vs. strengthening with age) is appropriate. Consider briefly mentioning whether executive function development or cognitive efficiency improvements might mediate these age effects, as this could inform future developmental research directions.

Comments 2.3-2.8:

The enhanced theoretical motivation for H2 and H3 in Section 1.1 is much clearer, with the distinction between cognition-based and affect-based factors now well-justified. I appreciate the explicit acknowledgment that the same tasks may capture different processes in humans vs. LLMs, and the removal of anthropomorphic language throughout improves scientific precision. The acknowledgment of potential confounds from linguistic complexity in the symmetry task and the cultural limitations of the U.S.-based sample are appropriate, with reasonable justification for maintaining sample consistency across studies. The expanded discussion of potential algorithmic improvements (fine-tuning, chain-of-thought prompting, contrastive learning, Direct Preference Optimization) is helpful, though appropriately presented as speculative. The inclusion of Study 2 with GPT-4o represents a significant strength that partially addresses concerns about generalizability across LLMs; the reference to Strachan et al. (2024) and calls for comparisons with fundamentally different architectures (Claude, Gemini, LLaMA) are appropriate, and the finding that additive bias persists across GPT-4 and GPT-4o suggests robustness within this model family.

ADDITIONAL COMMENTS

Study 2 Design:

The fully crossed 2x2x2 design in Study 2 is an improvement, and it also directly addresses Reviewer #3's concern about experimental design. This allows for proper testing of interactions between solution efficiency and valence of instruction, providing a more comprehensive understanding of the factors influencing additive bias. The improved accuracy rates in Study 2, particularly for GPT-4o's grid-based outputs in the symmetry task strengthen the interpretability of the findings, and the high consistency (in classifying solution strategies across response formats is reassuring. Also, the finding that instruction valence affects humans in the summary task but not the symmetry task in Study 2 is interesting and well-discussed. The interpretation that affective connotations may be more impactful in linguistic than spatial

tasks is reasonable and opens interesting avenues for future research.

MINOR SUGGESTIONS

1. Consider adding a brief summary table comparing key findings across Study 1 and Study 2 to help readers track the evolution and consistency of effects.
2. In the discussion of temperature settings (responses to Reviewer #3, comment 3.7), you appropriately acknowledge the lack of empirical validation. Consider briefly noting whether sensitivity analyses with different temperature values might be valuable future work.
3. The discussion could benefit from a brief comment on whether the stronger additive bias in GPT models might actually be adaptive in certain contexts (e.g., when users prefer elaboration or detail over minimalism).

RECOMMENDATION

The manuscript has been substantially improved through the addition of Study 2 and the comprehensive responses to reviewer concerns. The theoretical framing is now clearer, limitations are appropriately acknowledged, and the implications are more carefully stated. I recommend acceptance pending minor revisions to address the small suggestions above.

Reviewer #3 (Remarks to the Author):

I have thoroughly reviewed all of the authors' responses to my comments and questions. I have also inspected the revised sections of the paper that address my comments and can confirm that the authors have addressed all of my concerns to my satisfaction. I have no further questions or comments.

RESPONSE LETTER

Reviewer #1 (Remarks to the Authors):

Summary

This paper aims to explore how the efficiency of solutions and the valence of instructions influence the choice between additive or subtractive strategies when humans and generative artificial intelligence (particularly GPT-4) solve problems. Through four pre-registered experiments, the study investigates 588 American participants and 680 iterations of GPT-4 across spatial and language tasks. The research finds that GPT-4 exhibits a stronger additive bias than humans under most conditions, showing a greater tendency to choose strategies that increase elements. Human participants are less inclined to select additive strategies when subtractive strategies are more efficient, whereas GPT-4 displays the opposite trend under the same conditions. Additionally, the valence of instructions significantly affects GPT-4's strategy choices but has no significant impact on humans. The study concludes that cognitive biases in GPT-4 are amplified in certain aspects, suggesting caution when utilizing such models in practical applications.

Strengths

Novel and Practically Relevant Research Topic: This paper focuses on the additive bias in problem-solving between humans and LLMs, filling a gap in existing research regarding cognitive biases in LLMs. It holds high academic value and practical significance.

Rigorous Methodological Design: The study employs four pre-registered experiments covering both spatial and language tasks, controlling for solution efficiency and the valence of instructions. The experimental design is comprehensive and replicable.

Adequate Sample Size: The study meets the expected sample sizes for both human participants and GPT-4 iterations, enhancing the statistical power and credibility of the results.

Weaknesses

1.1 Task Comprehension Difficulties Affect Data Quality: Some human participants struggled to understand the instructions in spatial tasks, resulting in the exclusion of a significant amount of data. This not only impacted the final sample size but may also introduce selection bias, affecting the representativeness of the results.

We agree with the reviewer that difficulties in task comprehension, particularly in the symmetry task of Study 1, present a valid concern regarding both data quality and the generalizability of our findings. We addressed this issue systematically in Study 2 by revising the task instructions for clarity and accessibility.

The rate of data exclusion in the symmetry task was lower in Study 2 (see Experiment 5, mean: 55.9% of valid trials after exclusion due to the equal box count and unequal text-based and grid-based solution) compared to Study 1 (see Experiment 1 and 3, mean: 38.8% valid trials after exclusion due to the equal box count). Exclusion frequencies, exclusion criteria and other adjustments are detailed in the Method section of Study 2 and the Supplementary Information (SI 1 for Study 1 and SI 3 for Study 2). Nonetheless, participant exclusions were still high, which is discussed as a limitation in the General Discussion.

see 5.3 Materials (Study 2), page 24

see 8.1 Strengths and Limitations of the Study, page 39

We acknowledge the potential for selection bias due to data exclusions and now explicitly discuss this limitation in the General Discussion section. We emphasize the importance of future research employing validated comprehension checks, for example, to reduce exclusions and improve representativeness.

see 8.1 Strengths and Limitations of the Study, page 39

We thank the reviewer for this valuable observation, which has led to several enhancements in study design, reporting, and interpretation.

1.2 Insufficient Accuracy in GPT-4's Task Solutions: GPT-4 showed lower accuracy in solving spatial and summarization tasks, especially those requiring the understanding of complex instructions and negations. This calls into question the significance of analyses based on solution biases, as the model's inability to accurately complete tasks may complicate the interpretation of biased results.

We fully agree with the reviewer that interpreting solution strategy preferences (i.e., addition vs. subtraction) depends on a certain level of task understanding and accuracy. To address this concern, we made several changes in Study 2. First, we revised the instructions for both the symmetry and summary tasks to be more precise and consistent. For example, we rephrased ambiguous phrases like “edit/improve the summary” to “change/improve the summary” and clearly stated the action of toggling instead of switching to avoid confusion. For an overview of the complete material, see also Supplementary Information (SI 2 for Study 1, SI 4 for Study 2).

see 5.3 Materials (Study 2), page 24

In Study 1, accuracy rates for human participants ranged from 80% to 97% across conditions, while GPT-4 achieved rates below 20% in most tasks. In Study 2, humans again showed high accuracy (70-95%), while GPT-4o's text-based outputs remained mostly inaccurate (< 15%). However, in the spatial (symmetry) task, GPT-4o's grid-based outputs reached significantly higher accuracy levels (61-73%), indicating that the response format (text-based vs. grid-based) plays a crucial role in the accuracy of the symmetry task. Although the exact final box counts varied substantially between response formats (31.3% agreement), the key aspect – classifying the solution strategy as additive or subtractive – was highly consistent across response formats (98.0%), providing a reliable basis for inference. For an overview of all accuracies of solutions, see also Table 8 (for Study 1; page 19) and Table 16 (for Study 2; page 33). Results discussed in the main text, therefore, refer to the text-based responses, as the results based on grid-based responses showed the same pattern.

see 5.4 Procedure and Measures, pages 26-27

Nonetheless, we agree that the failures of GPT models to consistently solve specific tasks highlight fundamental differences between human and model reasoning. We now discuss these limitations more thoroughly in the General Discussion section. This also led us to refine our interpretation. Instead of attributing all additive preferences to “bias” in the cognitive sense, we now distinguish between potential bias-driven tendencies and structural output patterns that may stem from architectural or training constraints. We thank the reviewer for prompting this clarification.

see 8.1 Strengths and Limitations of the Study, pages 40-41

1.3 Inadequate Operationalization of Instruction Valence: Although the study attempted to manipulate the valence of instructions by changing verbs (e.g., "change" vs. "improve"), the results showed no significant impact on humans. This may be due to the manipulation of valence not being strong or specific enough to elicit the intended emotional responses effectively.

We thank the reviewer for this valuable comment. In the originally submitted study (Study 1), we indeed observed no significant effect of instruction valence on the strategy use of human participants.

As the reviewer rightly points out, this may be due to a too subtle manipulation of valence, especially in abstract problem contexts. Based on this feedback, we designed and conducted a follow-up study (Study 2), which implemented a revised and more consistent valence manipulation across both task types (symmetry and summary tasks).

see 4. Discussion (Study 1), pages 20-21

see 5.3 Materials (Study 2), page 24: “To strengthen the manipulation, the goal verb was included four times in the instructions, compared to once in the symmetry task and twice in the summary task in Study 1.”

In Study 2, the verbs “change” (neutral) and “improve” (positive) were more consistently integrated throughout the instruction texts, appearing at multiple points to reinforce the affective cue (e.g., “your task is to change/improve the pattern,” “describe how you change/improve the pattern,” and “as few changes/improvements as possible”). This adjustment aimed to increase the salience and perceived relevance of the valence manipulation. Again, for an overview of the complete material, see also Supplementary Information (SI 2 for Study 1, SI 4 for Study 2).

The new data revealed a significant effect of instruction valence on human participants’ choices in the summary task, but not in the symmetry task. This finding suggests that the revised manipulation, which used consistent and more salient valence framing, was indeed effective in one of the two problem types. We have updated the results section to reflect this pattern and now discuss in more detail why the effect may have emerged in the linguistic task but not in the spatial task. In particular, we consider that the affective connotations of verbs like “improve“ may be more impactful in linguistic tasks than in visually structured spatial tasks. We further address this asymmetry in the revised discussion and suggest possible ways to strengthen future valence manipulations (e.g., richer context, explicit emotion induction, or pre-tested valence measures).

see 7. Discussion (Study 2), page 35

see 8. General Discussion, page 37

see 8.3 Considerations for Future Research, page 43

1.4 Limited Cultural Diversity Restricts Generalizability: Participants were exclusively from the United States, ignoring how different cultural backgrounds might influence additive biases. This limits the applicability and generalizability of the findings on a global scale.

We have explicitly acknowledged the limited cultural scope of our sample in the limitations section of the revised manuscript.

While we agree that future research should investigate cultural differences in additive biases, especially given findings like those by Juvrud et al. (2024), we chose to use a U.S.-based sample throughout all studies to maintain internal consistency. Specifically, replicating the effects seen in Study 1 within Study 2 was a crucial part of our research design. Introducing more cultural variability at that stage would have made cross-study comparisons more difficult to interpret. Nonetheless, we agree that cross-cultural replication is an essential direction for future research.

see 8.1 Strengths and Limitations of the Study, page 39

see 8.2 Constraints on Generality, page 41

see 8.3 Considerations for Future Research, page 44

1.5 Lack of Exploration into LLMs' Internal Processes: While the study suggests that GPT-4 may possess cognitive mechanisms favoring addition, it lacks an in-depth analysis of the model's internal workings, such as the application of chain-of-thought prompting methods. This omission prevents a detailed understanding of the specific reasons behind the formation of biases in LLMs.

We thank the reviewer for this suggestion. We completely agree that a more detailed investigation of

LLM-internal processes is essential for understanding the mechanisms behind cognitive biases in these models.

In response, we have revised the relevant section of the manuscript to better acknowledge this limitation and to discuss possible future directions. Specifically, we now emphasize that applying, for example, chain-of-thought prompting in future research could help determine whether the additive bias results from a lack of reasoning depth, the structure of the training data, or other intrinsic properties of the model. We see this as a valuable methodological extension that would allow for more transparent and psychologically grounded comparisons between humans and LLMs.

see 8.1 Strengths and Limitations of the Study, pages 40-41

see 8.3 Considerations for Future Research, page 42

Reviewer #2 (Remarks to the Authors):

Review of “Influence of Solution Efficiency and Valence of Instruction on Additive and Subtractive Solution: Strategies in Humans and GPT-4” by Uhler et al.

OVERALL ASSESSMENT

The study examines additive and subtractive strategies in humans and GPT-4 in problem-solving tasks, taking into account solution efficiency and instruction valence. Both humans and GPT-4 showed the presence of addition bias (AB). Importantly, GPT-4 had a stronger additive bias, especially in the positive valence condition. Also, GPT-4’s decision-making process (DM) was less flexible than that of human participants, whose decisions were overall more efficient with regard to the outcome and less prone to the addition bias alone.

The study is quite timely and relevant as it informs our understanding of the mechanisms underlying AI performance as a potential aid to human DM. Notably, the paper extends previously reported data on the addition bias in LLMs by using both spatial and linguistic tasks and examining the contribution of cognitive and affective factors (i.e., solution efficiency and instruction valence), which expands the report’s theoretical value. The paper is well-written, the design is very elegant, and the statistical analyses are appropriate. I recommend minor revisions based on my comments provided below.

COMMENTS

2.1 Most of my comments focus on important limitations that were not mentioned or that were not discussed in enough depth. However, one major comment has to do with the **overall theoretical importance** of the paper. I admit this may necessitate a substantial degree of rewriting, so I leave it to the editor to decide if this is necessary before the paper can be accepted for publication. Nevertheless, my view is that any paper comparing AI performance to that of humans needs to explain how the reported data potentially answer one of the following general research questions, so I would prefer a deeper discussion of these:

- A. How accurately/to what degree does the training corpus used by different LLMs reflect human cognitive biases (and cognitive processes in general)? For example, some biases may have stronger sources in discourse rather than written texts. Similarly, the training corpora may have subsections that are suited better/worse to answer this question.
- B. What does analysis of LLM’s performance tell us about human cognition?
- C. What does analysis of LLM’s performance tell us about LLM’s “cognition”?
- D. How can LLM algorithms, etc., be improved in the future?

We thank the reviewer for encouraging us to clarify the theoretical basis of our work. We revised the manuscript to carefully explain what our comparisons of humans with GPT-4 (Study 1) or GPT-4o (Study 2) can and cannot reveal about A-D:

(A) Reflection of human biases in training corpora:

In the introduction, we now explicitly mention that there might be “several ‘entry points’ through which human regularities and biases can influence AI models, including (1) biased or unrepresentative training data and human annotations embedding social and linguistic regularities, (2) design and parameter choices shaped by model engineers’ implicit biases, and (3) insufficient awareness or negligence regarding fairness during model development (Osborne et al., 2023)” (see 1.3 Performance and Biases of LLMs: Previous Psychological Research, page 5). We cite Osborne et al. to place our work within the literature on the path from biased humans to biased data to biased models. Within this context, our results show that addition bias, an empirically observed human tendency (Adams et al., 2021), also appears in model outputs across two generations of models (GPT-4 and GPT-4o) and two task domains. We avoid claiming equivalence regarding internal processes and instead interpret the

consistent results across studies as reproducibility aligned with the bias transmission pathways identified by Osborne et al.

see 8.1 Strengths and Limitations of the Study, pages 40-41
see 8.3 Considerations for Future Research, page 42

(B) What LLM performance indicates about human cognition:

We adopt a cautious approach. Our design compares observable outputs generated under matched instructions and does not imply conclusions about process-level human cognition. Consequently, as noted in the General Discussion, similarities in strategy choice between humans and LLMs do not necessarily suggest shared mechanisms. The practical takeaway remains: LLM outputs are becoming increasingly integrated into human workflows and can influence human reasoning by promoting additive transformation-based solutions. We emphasise this applied influence, the impact on human decision-making, without drawing mechanistic inferences about human cognition.

see 8.1 Strengths and Limitations of the Study, pages 40-41

(C) What LLM performance indicates about LLM “cognition”:

As clarified in the Strengths & Limitations section, our studies do not examine internal computations. Investigating LLM “cognition” would require different methods, such as chain-of-thought protocols, representational or embedding analyses, or interventions in decoding or training processes. Therefore, we interpret the observed LLM patterns as statistical regularities in the responses to our prompts rather than evidence of internal reasoning comparable to human cognition. This limitation is now explicitly acknowledged in the manuscript.

see 8.1 Strengths and Limitations of the Study, pages 40-41

(D) Implications for improving models:

Algorithmic mitigation falls outside our empirical scope. Still, to acknowledge potential practical relevance, we briefly mention in the Considerations for Future Research section: “To reduce biases in problem-solving, such as the observed addition bias – whether termed a bias or seen as a reflection of patterns – future research could focus on fine-tuning LLMs. For instance, using datasets that highlight effective subtractive strategies or minimal interventions, or employing contrastive learning approaches that favor parsimonious over excessive changes (e.g., methods like Direct Preference Optimization; Rafailov et al., 2023) to align LLMs with human preferences. We present these as hypotheses for future technical research, not as definitive solutions. Systematic studies are necessary to evaluate the effectiveness of these adjustments.” (see 8.3 Considerations for Future Research, page 44).

OTHER COMMENTS

2.2 p. 3. The para starting with "Both..." needs a bit more work as it leaves the reader with little understanding of what it may mean that 9–10-year-olds use subtractive transformations less than adults. May it mean that what children are lacking is efficiency, which is sometimes achieved via subtraction instead of addition? May it also mean that addition bias is not enhanced by experience? Can it be that it becomes weaker with age instead, and what would that mean?

We thank the reviewer for this comment. We agree that our original phrasing did not fully explore the implications of developmental findings. We have now moved the paragraph surrounding the findings of Juvrud et al. (2024) to the General Discussion section and revised the paragraph to include alternative interpretations of Juvrud et al.’s results, such as the bias potentially diminishing or strengthening with age, and the possible influence of cognitive efficiency. This revised version better captures the open nature of this question and highlights the need for further developmental research.

see 8.3 Considerations for Future Research, page 44

2.3 Theoretical motivation of the task selection: Apart from AB, the inclusion of H2 and H3 on p. 6 is weakly motivated. Please expand Intro to offer stronger motivation to include these parameters.

We agree that the theoretical basis for including H2 (solution efficiency) and H3 (valence of instruction) should be more clearly outlined in the Introduction. We have therefore revised Section 1.1 to better highlight the reasoning behind these hypotheses, with a focus on cognition-based factors for H2 and affect-based factors for H3. Specifically, we now stress that manipulating solution efficiency helps test the flexibility of reasoning within structural constraints, while the valence manipulation explores the impact of affective framing on strategy choice. These updates clarify why both variables are essential for examining addition bias.

see 1.1 The Origins and Implications of Addition Bias, pages 3-4

2.4 Experimental tasks and the comparability between LLM's and human DM or problem solving: GPT-4's failure to produce accurate solutions in spatial tasks may reflect limitations in understanding abstract concepts rather than a general property of DM both in humans and LLMs. In other words, the DM process may be intrinsically different in AI and inhumans, and the same experimental task may be capturing somewhat different "cognitive" processes. In other words, while authors use the terms decision-making and problem solving when talking about human and AI performance, these may well be very different "cognitive" processes.

We thank the reviewer for raising this important conceptual point. We fully agree that although the same tasks are used for both agents, the internal mechanisms that generate human and AI responses are likely to differ significantly. In the revised manuscript, we now clearly address this limitation by explaining that our comparisons focus on functional-level output patterns rather than shared cognitive architecture.

see 8.1 Strengths and Limitations of the Study, pages 40-41

We also emphasize that terms like "decision-making" or "problem-solving" are used in a broad descriptive sense and should not imply identical processing mechanisms in humans and LLMs. Consequently, we have removed anthropomorphic language throughout the manuscript to better reflect this distinction.

2.5 Linguistic task: The symmetry task conditions have different levels of linguistic complexity, which may have led to comprehension problems rather than a modulation of AB in human participants. This is a potential confound that should be acknowledged as a limitation and a potential future direction. Same is true re. GPT-4's reliance on training corpus rather than problem-solving tendencies per se. This leads to another question: How accurately does the training corpus represent human AB? Future studies could assess whether modifications in training data or algorithmic adjustments reduce these biases.

We fully agree that differences in linguistic complexity in the symmetry task conditions may have affected task comprehension and potentially confounded the manipulation of solution efficiency in human participants. Similarly, the GPT model's output probably reflects patterns in its training data rather than problem-solving in a psychological sense. We have now explicitly addressed both limitations in the revised manuscript and proposed future directions to more precisely disentangle these effects.

see 8.3 Considerations for Future Research, page 42

see 8.3 Considerations for Future Research, page 43

2.6 Cultural constraints: The human sample was limited to U.S.-based participants, restricting the study's cultural generalizability. This is especially important in light of the findings reported in Juvrud et al. (2024) showing substantial cultural differences in the attribution of AB.

see our answer to Reviewer #1 „1.4 Limited Cultural Diversity Restricts Generalizability“

see 8.1 Strengths and Limitations of the Study, page 39

see 8.2 Constraints on Generality, page 41

see 8.3 Considerations for Future Research, page 44

2.7 Implications for AI Development: It is not particularly clear to me what algorithm or training corpus adjustments would make LLMs more optimal/efficient in terms of AB.

We appreciate this important observation. Although the current study was not designed to directly identify the mechanisms within the algorithms or training data that cause or reinforce additive bias, we agree that this is a key question for future research and AI development.

In the revised manuscript, we now detail how targeted fine-tuning, curated training data emphasizing subtractive reasoning, or prompting strategies like chain-of-thought reasoning, may help reduce the addition bias. Additionally, approaches like contrastive learning or exposure to tasks explicitly rewarding minimal change solutions could promote more balanced strategy use (e.g., methods like Direct Preference Optimization (Rafailov et al., 2023)). We recognize the speculative nature of these suggestions and highlight the need for systematic experimental variation in future LLM studies to evaluate their effectiveness.

see 8.3 Considerations for Future Research, page 44

2.8 Using other LLMs: Some recent studies showed important differences in how individual LLMs represent/reflect human biases (e.g., Strachan, J. W., Albergo, D., Borghini, G., Pansardi, O., Scaliti, E., Gupta, S.,... & Becchio, C. (2024). Testing theory of mind in large language models and humans. *Nature Human Behaviour*, 1-11.) It may well be that there are differences in how AB is represented in different LLMs as well. This would improve the AI generalizability of the reported data and offer a more nuanced understanding of AI's problem-solving strategies.

We thank the reviewer for this valuable point and the interesting literature reference. We fully agree that comparing different LLMs is essential to understanding the generalizability of cognitive biases across models, especially considering the findings of Strachan et al. (2024).

To address this, we

- discuss the importance of comparing models with different architectures: “Future research should extend such comparisons to fundamentally different architectures (e.g., Claude, Gemini, LLaMA). Recent evidence indicates that the reflection of human biases may vary considerably between model families. For example, Strachan et al. (2024) demonstrated notable differences between GPT and LLaMA2 models in how they exhibit and operationalize theory-of-mind reasoning.” (see 8.2 Constraints on Generality, page 42)
- included a second study using GPT-4o (see Study 2), the successor to GPT-4 (see Study 1), which differs in architecture and capabilities, especially in multimodal reasoning and efficiency (see 1.3 Performance and Biases of LLMs: Previous Psychological Research, pages 4-6). Using GPT-4o in Study 2 helped us to determine if the observed addition bias remains consistent across model versions. Results from Study 2 show that the overall tendency toward additive strategies persists in GPT-4o, indicating the bias's stability across related LLMs. We now explicitly highlight this in the revised manuscript as a strength of our design and as an initial step toward broader model comparisons.

Reviewer #3 (Remarks to the Authors):

Review: Influence of Solution Efficiency and Valence of Instruction on Additive and Subtractive Solution Strategies in Humans and GPT-4

I have some comments, mainly related to two main points: i) statistical analysis and ii) experimental design/setup. Further comments to minor points will follow (iii). Considering all my comments, I recommend that the authors make major revisions to the manuscript.

Major points

3.1 Statistical analysis: At first, I wondered why the authors did not incorporate random effects in their statistical models, e.g. via `glmer()` in the `{lme4}` R package, to account for interindividual variability between participants (or even GPT-4 iterations). However, after inspecting the provided code and data, I have to assume that each participant only did one experimental trial. This, however, is neither mentioned in Section 2.2 (Participants and Data Generation) nor Section 2.4 (Procedure and Measures). If it really is the case that each participant only provided one data point, then inclusion of participant ID as a random intercept indeed does not make any sense. However, the authors should mention this fact (= 1 data point per participant) in the paper.

This is correct; each human participant contributed a single data point in the experiment. Similarly, each GPT iteration counted as one data point. Therefore, including participant ID as a random effect would not be statistically meaningful. We have now explicitly clarified this in Study 1 in Section 2.2 (Participants and Data Generation) and in Study 2 in Section 5.2 by adding the following sentence: “Each human participant, as well as each GPT-4/GPT-4o iteration, contributed a single response in the experiment. Thus, the dataset contains one observation per case.” (see 2.4 Procedure and Measures, Study 1, page 13; see 5.4 Procedure and Measures, Study 2, page 22)

However, there is one point that the authors should definitely consider in their statistical analysis. They always perform a logistic regression analysis (with the predicting structure `agent*condition`) first, followed by a post-hoc analysis (effect of condition, separate for both agents). However, this post-hoc analysis never corrects for multiple comparisons. If the authors believe that no correction is necessary here, they would have to provide sufficient justification.

We appreciate the reviewer’s attention to the issue of multiple comparisons. We did not apply a correction for multiple testing in the post-hoc analyses because these comparisons were pre-specified and theory-driven, as outlined in our preregistration (<https://osf.io/6pkwb>). Within each experiment of Study 1, we tested a single interaction model with two factors (agent x condition), and followed up when the interaction was significant. Furthermore, the number of planned comparisons was strictly limited (two per experiment), and they were structurally derived from the experimental design and hypotheses (H2 and H3). We therefore consider the risk of inflated type I error to be minimal and controlled. If necessary, we are happy to transparently report adjusted p-values (e.g., Bonferroni), but we believe that the current approach aligns with best practice.

For Study 2, the same principle applies. The analyses were limited to a predefined set of hypothesis-driven tests: a chi-square test for overall addition bias (H1), a chi-square test of homogeneity to examine differences between agents (RQ), and two logistic regression models per experiment to test H2 (solution efficiency) and H3 (valence of instruction), including their interactions with agent. Post-hoc tests were only conducted when the interaction terms were significant and followed a pre-specified hierarchical logic (i.e., breaking down three-way interactions into two-way interactions and simple main effects). Given the small and theoretically motivated set of planned comparisons, we again consider the likelihood of inflation of Type I error to be low, and the analytical approach appropriately controlled.

see 2.4 Analytic Strategy (Study 1), page 15: “All models, comparisons, and post-hoc analyses were pre-registered, following the experimental design and specific directional hypotheses. Therefore, no correction for multiple testing was used.”

see 4.4 Analytic Strategy (Study 2), page 28: “As in Study 1, all models, comparisons, and post-hoc analyses were pre-registered, following the experimental design and specific directional hypotheses.”

3.2 Experimental design/setup: I wonder why the authors decided to set up the various experiments the way they did. Exp. 1 and 3 vary “Solution efficiency”, Exp. 2 and 4 vary “Valence of instruction”. I see no reason why these two factors were spread across experiments and not fully crossed in one experiment to compare the effects of the two factors within a design and possibly also test for potential interactions (efficiency*valence). The full analysis would then also include the agent (efficiency*valence*agent), which would mean that a potential 3-way interaction would have to be interpreted, but this would be somewhat straightforward, e.g. a different effect of the interaction “efficiency:instruction” depending on the agent.

When initially designing the experiments (now Study 1 in the revised manuscript, comprising Experiments 1-4), we conceptualized solution efficiency and valence of instruction as two distinct and independent factors influencing addition bias. Our primary objective was to investigate the influence of each of these two manipulations separately, and to determine if potential patterns of influence would differ by agent (humans vs. GPT-4). Consequently, our original hypotheses and research questions were not focused on exploring potential interactions between solution efficiency and valence of instruction.

see Study 1: Addition Bias in Humans vs. GPT-4, starting page 8

We agree that a fully crossed design, including solution efficiency and valence of instruction (alongside agent), would allow for a more comprehensive analysis and the examination of possible two-way and three-way interactions.

We are pleased to report that we have conducted a follow-up study using a 2 (solution efficiency: addition and subtraction equally efficient vs. subtraction more efficient) x 2 (valence of instruction: neutral vs. positive) x 2 (agent: human vs. GPT-4o) between-subjects design.

See 4. Discussion (Study 2), *Limitations and transition to Study 2*, pages 20-21
see Study 2: Addition Bias in Humans vs. GPT-4o, starting page 21

The additional Experiments 5 (symmetry task) and 6 (summary task) are now included in the revised manuscript (see Study 2: Addition Bias in Humans vs. GPT-4o), enabling us to test exactly the interactions the reviewer pointed out. We appreciate the reviewer’s suggestion and hope the new study enhances the overall contribution of the paper.

I also see problems in the ‘naturalness’ of the tasks, which the authors themselves also address to some point. This applies to both tasks (symmetry & summary). In the symmetry task, the problem is that the human participants (but also GPT-4) had to solve a spatial task in a linguistic way that might have been easy to solve interactively (for example via an interface where it is actually possible to switch Xs on and off). This means that the authors cannot be sure whether a particular participant (or GPT-4) really could not solve the primary symmetry task or simply failed due to the linguistic description of their solution (their ‘secondary’ task). For the summary task, the ‘naturalness problem’ is somewhat different. Each of us probably had to lengthen or (probably more often) shorten a text – a scenario that many of the participants might know or might have experienced themselves in the past. In the experiment, however, a text had to be changed in such a way that a certain range of word counts was not reached (equally efficient: 64 to 88 words; subtraction more efficient: 70 to

94 words). The extremely high rates of inaccurate solutions (especially the catastrophic figures for GPT-4 in the symmetry task) reported in Table 2 could also be interpreted as an expression of a lack of ‘naturalness’ in both tasks. It is difficult to give a clear recommendation here except to repeat the study with more natural tasks. With the current design, I can only suggest that the caveats be addressed very clearly.

We thank the reviewer for raising this important concern about task naturalness.

We fully agree that the ecological validity of experimental tasks is essential, especially when studying cognitive strategies. In our current studies, we intentionally replicated the task structure from Study 1 in Study 2 to maintain consistency and comparability across both experiments.

Regarding the symmetry task, we acknowledge that solving a spatial pattern problem through linguistic output is somewhat artificial. To address this, we revised the task instructions in Study 2 to provide clearer guidance, aiming to reduce ambiguity about how participants can describe their solutions.

For the summary task, we agree that the solution-efficiency manipulation using target word ranges may seem somewhat artificial. However, we note that real-world writing tasks, such as abstracts, applications, and assignments, often involve constraints on word count, making the trade-off between shortening and extending text a plausible, albeit simplified, decision scenario.

Our main goal in these studies was to test basic cognitive tendencies in problem reformulation under controlled conditions, similar to previous work by Adams et al. (2021), where tasks like modifying a minigolf course or stabilizing a LEGO tower were only loosely representative of real-world decisions.

We have clearly addressed these limitations in the revised manuscript and strongly encourage future research to explore the same mechanisms using more ecologically valid, interactive, or domain-specific tasks.

see **8.1 Strengths and Limitations of the Study, pages 39-40**

Minor points

3.3. Testing “anchoring bias”: On page 19, the authors discuss an apparently 'irrational behavior' of GPT-4 and cite a potential “anchoring bias” as an explanation. The authors could test this explanation directly if they agree with my below suggestion. The problem identified is that “the baseline number of filled fields is smaller” in the “subtraction more efficient” condition. The authors could easily test whether this explanation is correct by choosing a subtraction-more-efficient condition that also has 6 activated fields (X), just like the equally-efficient condition. To achieve this, one would have to setup a starting grid like (for example) this:

```
A B C D
1 [X] [] [] []
2 [] [X] [X] []
3 [] [X] [X] []
4 [] [] [X] []
```

With a starting grid like this, subtraction is still more efficient because two X (A1, C4) would have to be removed to achieve symmetry. A solution involving addition would at least require three additions (D1, A4, D4) and one subtraction (C4). If the lower number of active fields indeed has an effect on GPT-4s performance, this should not be the case for a starting grid as illustrated above. For testing this, only additional

runs for GPT-4 would have to be collected.

We thank the reviewer for this constructive and insightful suggestion. The reviewer's proposed grid would alter the initial setup so that both the "equal efficiency" and "subtraction more efficient" conditions start with the same number of filled fields. However, this change would fundamentally alter the task logic: in the proposed arrangement, even additive strategies would necessarily involve at least one subtractive transformation.

We improved the original task setup from Study 1 to Study 2 by clarifying instructions and removing linguistic ambiguity. This change was successful: GPT-4o achieved relatively high accuracy, even in conditions that included "subtraction more efficient," when responses were generated in the grid-based format (62% and 65%; see Table 16, page 33).

Nevertheless, we agree that anchoring effects in symmetry-based reasoning tasks represent an important theoretical avenue. We now explicitly mention in the General Discussion that future research should examine potential anchoring influences by independently manipulating the number of initially filled fields and the efficiency structure, as proposed by the reviewer.

see 8.1 Strengths and Limitations of the Study, page 40

3.4. Adams et al. (2021): On page 3, the authors write: "Since the initial investigation of addition bias, studies showed that its effects are anchored in various domains that extend far beyond the anecdotal evidence [my emphasis] presented by Adams et al. (2021)." I know which parts of Adams et al. (2021) the authors refer to, but the sentence might be interpreted in a way that the authors certainly did not intend. The experimental evidence provided by Adams et al. (2021) is certainly not "anecdotal", only some of the domains Adams et al. (2021) provide citations to (schedules, institutional red tape, damaging effects on the planet) could be considered "anecdotal".

We thank the reviewer for pointing out the ambiguity in our original wording. Our intention was not to describe the experimental evidence in Adams et al. (2021) as anecdotal. Instead, we referred to the illustrative real-world examples used by Adams et al. to frame their study, as correctly stated ("schedules, institutional red tape, damaging effects on the planet"). Our point was that subsequent research has expanded the investigation of addition bias far beyond these initial domains, demonstrating its relevance across multiple contexts. We have revised the sentence to clarify this.

Revised sentence: "Since the initial investigation of addition bias, studies showed that its effects are anchored in various domains that extend far beyond **the empirical evidence and the illustrative real-world examples** presented by Adams et al. (2021)."

see 1.1 The Origins and Implications of Addition Bias, page 4

3.5 Annotation: How many people annotated the answers given in the symmetry task? Was the annotation checked for inter-rater reliability? Please address this in the text.

We thank the reviewer for this valuable comment. In Study 1, annotation was conducted by a single, well-trained rater. We recognize that this limits the ability to measure rater agreement. To address this issue, Study 2 involved two independent coders, and inter-rater reliability was assessed (Cohen's $\kappa = .911$). We have clarified this in the revised manuscript.

see 2.4, Procedure and Measures (Study 1), page 14

see 5.4, Procedure and Measures (Study 2), page 26

3.6 Power calculation (p. 7, right below section header 2.2): The preregistration by the authors provides much more detail about the power calculations. Please refer to the preregistration here to allow readers easy reference.

We thank the reviewer for pointing this out. We have now added a direct reference to the preregistration (both in Study 1 and Study 2), including the full details of our power analysis, to enhance transparency and facilitate easy access.

see **2.2 Participants and Data Generation (Study 1), page 8**
see **5.2 Participants and Data Generation (Study 2), pages 21-22**

3.7 Temperature Setting: Is there any citation available that “a temperature setting of 0.7” for GPT-4 simulates “human-like answer variability”? (p. 7)

We thank the reviewer for this comment. We agree that there is currently no published empirical evidence explicitly validating a temperature setting of the default values 0.7 (Study 1) or 1.0 (Study 2) as a way to simulate „human-like answer variability.“ We have revised the manuscript to reflect this, now referring more cautiously to this parameter choice as a practical heuristic based on OpenAI API documentation rather than an empirically supported standard.

see **2.2 Participants and Data Generation (Study 1), page 8**
see **2.4 Procedures and Measures (Study 1), page 13**

see **5.2 Participants and Data Generation (Study 2), page 22**
see **5.4 Procedures and Measures (Study 2), page 26**

RESPONSE LETTER

REVIEWERS' COMMENTS:

Reviewer #2 (Remarks to the Author):

OVERALL ASSESSMENT

Thank you for your comprehensive responses to my comments and for conducting Study 2, which significantly strengthens the manuscript. I appreciate the substantial revisions made throughout the paper, particularly the improved theoretical framing and discussion of limitations. Below are my comments on the revised manuscript.

REVISION 1 COMMENTS

Comment 2.1:

I am satisfied with how you have addressed the theoretical positioning of your work, particularly regarding points A-D. The clarifications about what your comparisons can and cannot reveal are much clearer now. The acknowledgment that similarities in output patterns do not imply shared mechanisms is appropriate and prevents overinterpretation of the findings. The discussion of bias transmission pathways (Osborne et al., 2023) provides valuable context.

However, I would encourage slightly more elaboration on point (A) regarding how the training corpus might differentially represent additive vs. subtractive strategies in text. For example, are there linguistic or discourse features that might make additive transformations more frequent or salient in written corpora? This could provide additional theoretical depth to explain why GPT models show stronger additive bias than humans.

We thank the reviewer for this additional suggestion and fully agree that the differential representation of additive versus subtractive transformations in written language warrants a more explicit discussion.

We have therefore expanded the manuscript to include concrete linguistic and corpus-based evidence showing that written language systematically overrepresents additive transformations. Specifically, in Section 1.3 Performance and Biases of LLMs: Previous Psychological Research (page 5), we now refer to work demonstrating that (i) LLMs assign higher cloze probabilities to additive than to subtractive continuations after verbs of change, and that change- and improvement-related verbs are more strongly associated with addition than subtraction in semantic embedding spaces (Winter et al., 2023), and (ii) frequency analyses of large text corpora show that words related to “add” and “more” occur more often than “remove” or “subtract” in both English and German (Wolfer, 2023).

Building on these findings, we argue that large-scale training corpora systematically prioritise additive discourse operations, providing a concrete corpus-linguistic mechanism for why GPT models may exhibit a more substantial additive bias than humans, beyond general bias transmission pathways (Osborne et al., 2023).

Comment 2.2:

The relocation and expansion of the discussion regarding Juvrud et al.'s (2024) developmental findings is an improvement. The acknowledgment of multiple possible interpretations (bias weakening vs. strengthening with age) is appropriate. Consider briefly mentioning whether executive function development or cognitive efficiency improvements might mediate these age effects, as this could inform future developmental research directions.

We appreciate the suggestion to link the developmental findings to executive function. We agree that these mechanisms offer a theoretically grounded way to interpret the reported age effects and their ambiguity. In the paragraph discussing Juvrud et al. (2024), Section 8.3 Considerations for Future

Research (page 46), we added a brief passage noting that improvements in executive functions – such as inhibitory control, working memory, and cognitive efficiency – may plausibly mediate age-related changes in additive bias. We explicitly frame this as a testable hypothesis for future studies rather than a definitive explanation.

Comments 2.3-2.8:

The enhanced theoretical motivation for H2 and H3 in Section 1.1 is much clearer, with the distinction between cognition-based and affect-based factors now well-justified. I appreciate the explicit acknowledgment that the same tasks may capture different processes in humans vs. LLMs, and the removal of anthropomorphic language throughout improves scientific precision. The acknowledgment of potential confounds from linguistic complexity in the symmetry task and the cultural limitations of the U.S.-based sample are appropriate, with reasonable justification for maintaining sample consistency across studies. The expanded discussion of potential algorithmic improvements (fine-tuning, chain-of-thought prompting, contrastive learning, Direct Preference Optimization) is helpful, though appropriately presented as speculative. The inclusion of Study 2 with GPT-4o represents a significant strength that partially addresses concerns about generalizability across LLMs; the reference to Strachan et al. (2024) and calls for comparisons with fundamentally different architectures (Claude, Gemini, LLaMA) are appropriate, and the finding that additive bias persists across GPT-4 and GPT-4o suggests robustness within this model family.

We thank the reviewer for the positive evaluation of the strengthened theoretical framing and the clarification of the distinction between cognition-based and affect-based hypotheses. We also appreciate the acknowledgement of improvements in the handling of linguistic confounds, cultural limitations, and model generalizability. We agree that the speculative discussion of algorithmic mechanisms is appropriate as a forward-looking perspective rather than a definitive account.

ADDITIONAL COMMENTS

Study 2 Design:

The fully crossed $2 \times 2 \times 2$ design in Study 2 is an improvement, and it also directly addresses Reviewer #3's concern about experimental design. This allows for proper testing of interactions between solution efficiency and valence of instruction, providing a more comprehensive understanding of the factors influencing additive bias. The improved accuracy rates in Study 2, particularly for GPT-4o's grid-based outputs in the symmetry task strengthen the interpretability of the findings, and the high consistency (in classifying solution strategies across response formats) is reassuring. Also, the finding that instruction valence affects humans in the summary task but not the symmetry task in Study 2 is interesting and well-discussed. The interpretation that affective connotations may be more impactful in linguistic than spatial tasks is reasonable and opens interesting avenues for future research.

We thank the reviewer for recognizing the methodological improvement achieved by the fully crossed $2 \times 2 \times 2$ design in Study 2 and for highlighting its value in addressing interaction effects and previous design limitations. We also appreciate the positive assessment of the improved accuracy and consistency of GPT-4o outputs.

MINOR SUGGESTIONS

1. Consider adding a brief summary table comparing key findings across Study 1 and Study 2 to help readers track the evolution and consistency of effects.

We agree that a comparative overview facilitates tracking the consistency of effects across studies. We would like to note that such a summary table was already included in the revised manuscript. The comparative overview is provided in Table 17, Section 8 (General Discussion) on page 38.

2. In the discussion of temperature settings (responses to Reviewer #3, comment 3.7), you appropriately acknowledge the lack of empirical validation. Consider briefly noting whether sensitivity analyses with different temperature values might be valuable future work.

We agree that systematic temperature-sensitivity analyses represent an interesting methodological extension and constitute valuable future work. In Section 8.3, Considerations for Future Research (page 44+45), we now explicitly note that future studies should conduct controlled sensitivity analyses across multiple temperature settings to assess the impact on addition bias.

3. The discussion could benefit from a brief comment on whether the stronger additive bias in GPT models might actually be adaptive in certain contexts (e.g., when users prefer elaboration or detail over minimalism).

We appreciate this perspective and agree that addition bias should not be interpreted as inherently maladaptive. Adding components to objects, situations, or ideas may be desirable in specific situations depending on user goals. We added a brief note to Section 8. General Discussion (page 40) stating that, in certain contexts, additive transformations may be preferred over minimalist or reductive solutions, particularly when enrichment, elaboration, or augmentation of existing structures is explicitly desired.

RECOMMENDATION

The manuscript has been substantially improved through the addition of Study 2 and the comprehensive responses to reviewer concerns. The theoretical framing is now clearer, limitations are appropriately acknowledged, and the implications are more carefully stated. I recommend acceptance pending minor revisions to address the small suggestions above.

Reviewer #3 (Remarks to the Author):

I have thoroughly reviewed all of the authors' responses to my comments and questions. I have also inspected the revised sections of the paper that address my comments and can confirm that the authors have addressed all of my concerns to my satisfaction. I have no further questions or comments.

We sincerely thank Reviewer #3 for the careful re-evaluation of the revised manuscript and for confirming that all concerns have been satisfactorily addressed.